# Rapid transport of deformation-tuned nanoparticles across biological hydrogels and cellular barriers

Miaorong Yu[1,2], Lu Xu[3], Falin Tian[4], Qian Su[2,4,5], Nan Zheng[3], Yiwei Yang[1,2], Jiuling Wang[2,4,5], Aohua Wang[1,2], Chunliu Zhu[1], Shiyan Guo[1], XinXin Zhang[1], Yong Gan[1,2], Xinghua Shi [2,4] & Huajian Gao[6]

To optimally penetrate biological hydrogels such as mucus and the tumor interstitial matrix, nanoparticles (NPs) require physicochemical properties that would typically preclude cellular uptake, resulting in inefficient drug delivery. Here, we demonstrate that (poly(lactic-co-glycolic acid) (PLGA) core)-(lipid shell) NPs with moderate rigidity display enhanced diffusivity through mucus compared with some synthetic mucus penetration particles (MPPs), achieving a mucosal and tumor penetrating capability superior to that of both their soft and hard counterparts. Orally administered semi-elastic NPs efficiently overcome multiple intestinal barriers, and result in increased bioavailability of doxorubicin (Dox) (up to 8 fold) compared to Dox solution. Molecular dynamics simulations and super-resolution microscopy reveal that the semi-elastic NPs deform into ellipsoids, which enables rotation-facilitated penetration. In contrast, rigid NPs cannot deform, and overly soft NPs are impeded by interactions with the hydrogel network. Modifying particle rigidity may improve the efficacy of NP-based drugs, and can be applicable to other barriers.

[1] Shanghai Institute of Materia Medica, Chinese Academy of Sciences, 201203 Shanghai, China. [2] University of Chinese Academy of Sciences, NO.19A Yuquan Road, 100049 Beijing, China. [3] School of Pharmacy, Shenyang Pharmaceutical University, 110016 Shenyang, China. [4] CAS Key Laboratory for Nanosystem and Hierarchy Fabrication, CAS Center for Excellence in Nanoscience, National Center for Nanoscience and Technology, Chinese Academy of Sciences, 100190 Beijing, China. [5] LNM, Institute of Mechanics, Chinese Academy of Sciences, 100190 Beijing, China. [6] School of Engineering, Brown University, Providence, RI 02912, USA. These authors contributed equally: Miaorong Yu, Lu Xu, Falin Tian. Correspondence and requests for materials should be addressed to Y.G. (email: ygan@simm.ac.cn) or to X.S. (email: shixh@nanoctr.cn) or to H.G. (email: huajian_Gao@brown.edu)

Biological hydrogels, such as the mucus lining all wet epithelia[1,2] and the tumor interstitial matrix[3], are substantial barriers for nanoparticle (NP)-based drug delivery systems[4]. Both mucus and the tumor interstitial matrix have a cross-linked mesh structure composed of protein fibers that traps NPs. Numerous studies have been performed to overcome these biological barriers either for mucosal drug delivery or for cancer therapy[5–7]. However, there is evidence indicating that the physicochemical properties required for NPs to efficiently penetrate hydrogels are typically unsuitable for cellular internalization[8,9], making drug delivery to tissues covered with biological hydrogels challenging. For instance, positively charged NPs are favorable for cellular uptake but are ill suited for penetrating biological hydrogels[10,11]. Decreasing the size of NPs is an effective way to facilitate hydrogel penetration; however, extremely small NPs (2–10 nm) are inefficiently endocytosed[12,13]. NPs modified with a high density of low-molecular-weight poly (ethylene glycol) (PEG) can be efficiently transported across mucus and the tumor interstitial matrix[14–16]. Nevertheless, these PEGylated NPs also display a pronounced reduction in their cellular uptake compared with bare NPs[17]. Therefore, simultaneously overcoming the biological hydrogel barrier and the cellular barrier has become a major challenge in the design of efficient drug delivery systems targeting tissues covered with biological hydrogels.

Inspired by findings that the elastic modulus could play a governing role in the translocation, biodistribution, and circulation of red blood cells (RBCs) and cancer cells[18,19], researchers have synthesized particles with different rigidities and have sought to address the complex effects of particle rigidity on biological fates[20–23]. Alexander et al. found that soft NPs were internalized by SUM159 cancer cells to a greater extent than hard NPs after 2 h of incubation, but this relationship was reversed after 4 h of incubation[24]. Merkel et al. showed that hard RBC-shaped particles displayed high levels of accumulation in the heart and lung, whereas soft particles displayed high levels of accumulation in the spleen[25]. In vitro studies have demonstrated that soft particles exhibit low levels of internalization into immune cells; this finding was supported by in vivo studies showing that soft particles circulated for longer periods than their hard counterparts[25,26]. However, the underlying mechanism responsible for why and how NP rigidity affects drug delivery processes remains largely unexplored. This task is not a trivial one because, in biological environments, visualizing the microscopic movement of NPs, particularly their rotation and deformation, requires high sensitivity and spatiotemporal resolution.

In this study, we systematically evaluate the effect of rigidity on transport capability of NPs in tissues covered with biological hydrogels using core-shell polymer-lipid NPs. Specifically, we adopt a microfluidic-based system to assemble core-shell poly (lactic-co-glycolic acid) (PLGA)-lipid NPs with different rigidities, which are tuned by varying the amount of interfacial water between the PLGA core and the lipid shell. We observe that NPs with intermediate rigidity (i.e., semi-elastic NPs) display superior diffusion through multiple intestinal and tumor barriers compared with NPs of high or low rigidity. Orally administered Dox-loaded semi-elastic NPs show increased bioavailability of doxorubicin (Dox) compared to Dox solution. Molecular dynamics (MD) simulations and super-resolution microscopy reveal that the semi-elastic, spherical NPs deform into ellipsoids within the complex hydrogel mesh structure, which ultimately induces rotation-facilitated fast diffusion. In contrast, soft NPs change their shapes excessively and irregularly, interspersing with the gel polymers and becoming confined within them; thus, they exhibit the lowest permeation.

## Results

**Characterizations of NPs of varying rigidities.** In this study, we developed a two-step approach to assemble core-shell PLGA-lipid NPs with different rigidities by tuning the amount of interfacial water between the PLGA core and the lipid shell: the NPs became increasingly soft as more interfacial water was added[27–29]. PLGA cores with various hydrodynamic diameters (50, 70, 90, 110, 130, and 160 nm) were first fabricated using a microfluidic-based system by modulating the flow rate. The bare PLGA cores were subsequently encapsulated in liposomes composed of egg phosphatidylcholine (EPC) using a microextruder device to obtain the final core-shell PLGA-lipid NPs, as shown schematically in Fig. 1a. We refer to the PLGA-lipid NPs using the notation $PLGA_x$-Lip-$F127_{5\%}$, where $PLGA_x$ indicates a PLGA core of diameter $x$ nm and $F127_{5\%}$ indicates 5 wt% content of Pluronic F127 (F127) on the lipid (Lip) shell. Hollow liposomes with and without F127 modification are referred to as Lip-$F127_{5\%}$ and Lip, respectively.

As shown in Supplementary Table 1, all NPs had similar hydrodynamic diameters (approximately 200 nm) and surface charges (approximately-5 mV). This enables us to explore how the rigidity of NPs differentially regulates their permeation ability in biological gels. To characterize the amounts of interfacial water inside the hybrid NPs, we used ethyl rhodamine B to visualize the volume occupied by the water encapsulated by the NPs according to a previously reported method[29]. The fluorescence intensity and therefore the amount of interfacial water of the NPs decreased as the hydrodynamic diameter of the PLGA cores increased (Fig. 1d). The core-shell structure of the NPs was further confirmed by cryogenic transmission electron microscopy (cryo-TEM; Fig. 1b), which demonstrated that the lipid shells completely covered the PLGA cores. For $PLGA_{160}$-Lip-$F127_{5\%}$, a thin water layer existed between the lipid shell and the PLGA core. For $PLGA_{70}$-Lip-$F127_{5\%}$, however, the interfacial water layer was moderate in size; smaller amounts of interfacial water would result in more rigid NPs[27,28].

AFM was then applied to validate the distinct rigidity of the NPs. The Young's modulus of NPs increased as the hydrodynamic diameter of the PLGA cores increased. The Young's modulus of $PLGA_{70}$-Lip-$F127_{5\%}$ NPs was approximately 50 MPa, which is 10-fold higher than that of the Lip-$F127_{5\%}$ NPs (5 MPa) and 2-fold lower than that of the $PLGA_{160}$-Lip-$F127_{5\%}$ NPs (110 MPa) (Fig. 1c). Next, the morphologies of three types of NPs (i.e., Lip-$F127_{5\%}$ (soft), $PLGA_{70}$-Lip-$F127_{5\%}$ (semi-elastic), and $PLGA_{160}$-Lip-$F127_{5\%}$ (hard)) were characterized using AFM-based nanomechanical mapping. The topographies of the NPs were typically spherical (Fig. 1e); however, as the force was increased from 500 pN to 50 nN, the NPs gradually deformed into different shapes (Fig. 1e). The soft NPs deformed irregularly, the semi-elastic NPs deformed into ellipsoidal particles, and the hard NPs displayed negligible deformation. These variations in deformation can be attributed to the varying rigidities of the NPs, which resulted from the different amounts of interfacial water encapsulated in the NPs. Finally, to evaluate the biological stability of the different PLGA-lipid NPs synthesized, NPs were incubated in simulated gastric fluid (SGF) and simulated intestinal fluid (SIF). All the NPs tested remained stable in the SGF and SIF for 6 h (Supplementary Note 1 and Supplementary Fig. 1).

**Particle tracking in mucus in vitro.** We then tracked the motion of NPs in freshly obtained, undiluted rat intestinal mucus, as a representative biological hydrogel. It was observed that the $PLGA_{70}$-Lip-$F127_{5\%}$ NPs readily exhibited the fastest diffusion in the mucus and displayed the largest diffusion areas of the NPs

tested (Fig. 2a). The time scale-dependent ensemble mean squared displacement (<MSD>) values of the NPs were quantified to evaluate the extent of impediment to particle transport (Fig. 2b). The rapid mobility of the $PLGA_{70}$-Lip-$F127_{5\%}$ (semi-elastic) NPs is reflected by their <MSD> values being markedly higher across all time scales than those of the Lip-$F127_{5\%}$ (soft) and $PLGA_{160}$-Lip-$F127_{5\%}$ (hard) NPs. On a time scale of 1 s, the <MSD> value of the $PLGA_{70}$-Lip-$F127_{5\%}$ (semi-elastic) NPs was approximately 97.5-fold, 11.0-fold, and 8.5-fold higher than those of the Lip, Lip-$F127_{5\%}$ (soft), and $PLGA_{160}$-Lip-$F127_{5\%}$ (hard) NPs (Supplementary Table 1), respectively. We also tracked the motion of NPs in poly(ethylene oxide) (PEO) hydrogels for comparison and obtained similar results (Supplementary Note 2 and Supplementary Fig. 2). In addition, we tracked the transport of mucus adhesive particles, and the results indicated that the observations and conclusions in this study were free of potential biases (Supplementary Note 3 and Supplementary Fig. 3).

Based on a previous study[30], we would expect unusually fast-moving NPs to be capable of penetrating mucus layers; thus, we characterized the distribution of individual particle effective

diffusivity ($D_{eff}$) values in mucus on a time scale of 1 s (Fig. 2c). We found that all $PLGA_{70}$-Lip-$F127_{5\%}$ (semi-elastic) NPs diffused relatively quickly, with $D_{eff}$ values exceeding 0.1 $\mu m^2$/s. In contrast, only 4.7% of the Lip-$F127_{5\%}$ (soft) NPs and 3.3% of the $PLGA_{160}$-Lip-$F127_{5\%}$ (hard) NPs exceeded this diffusion speed.

**Cellular uptake.** For in vitro studies of the cellular uptake of NPs, we first used the HT29-MTX-E12 (E12) cell line, which consists of a secreted mucus layer and an absorptive epithelial cell layer, to mimic mucosal tissue[8]. We also removed the secreted mucus layer using N-acetylcysteine (NAC)[31] for comparison. Both pre-treated and non-treated cells were incubated with Lip-$F127_{5\%}$ (soft), $PLGA_{160}$-Lip-$F127_{5\%}$ (hard), and $PLGA_{70}$-Lip-$F127_{5\%}$ (semi-elastic) NPs for 2 h to quantitatively study the cellular uptake of NPs. As shown in Fig. 3a, the semi-elastic NPs exhibited the best cellular internalization capability among the tested samples. Notably, for the semi-elastic NPs, there was no significant difference in uptake between the groups with and without mucus pre-removal, whereas the soft and hard NPs exhibited significantly lower uptake when mucus was present. We also

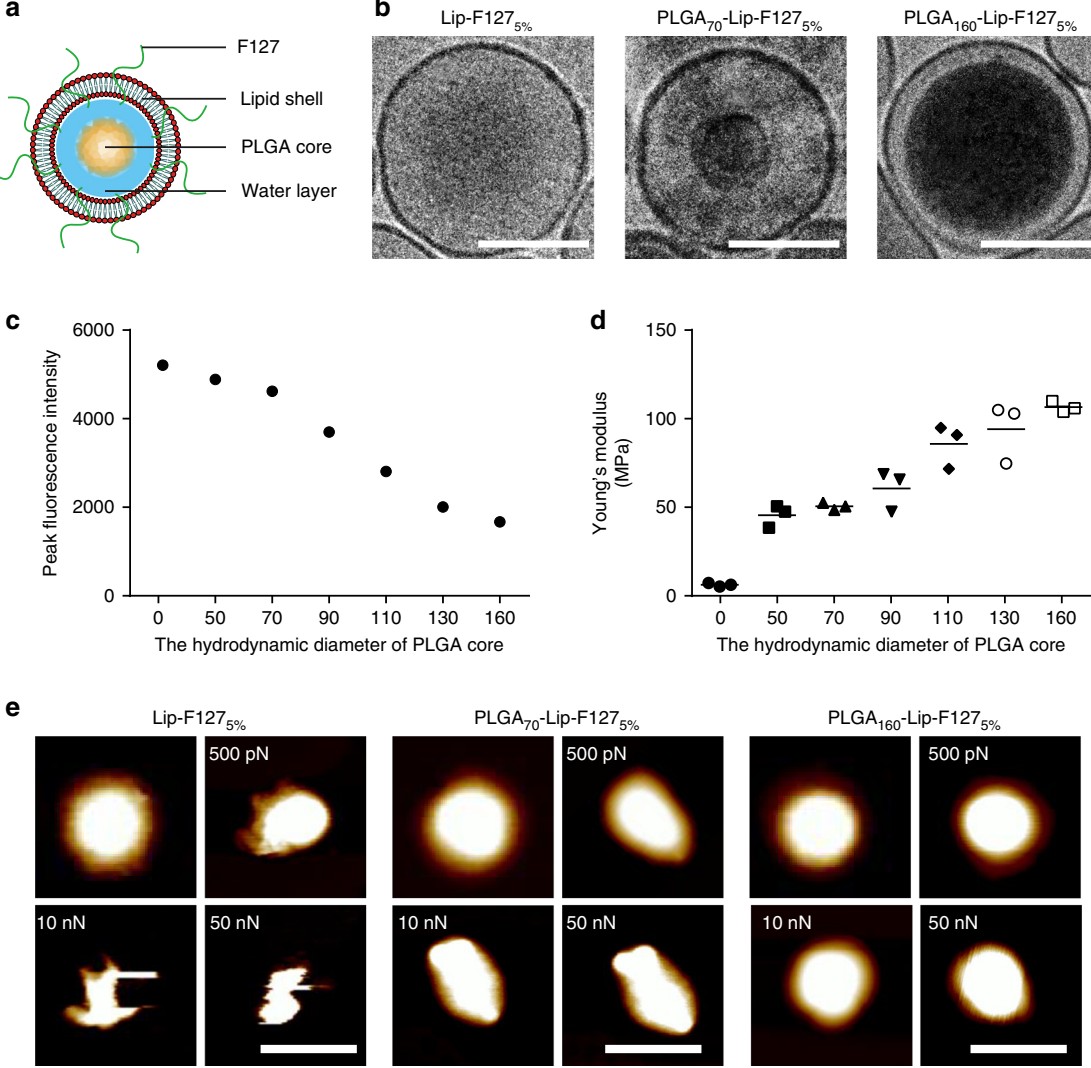

**Fig. 1** NP characterization. **a** A schematic diagram of a core-shell PLGA-lipid NP. **b** Cryo-TEM images of a Lip-$F127_{5\%}$ NP (left), a $PLGA_{70}$-Lip-$F127_{5\%}$ NP (middle), and a $PLGA_{160}$-Lip-$F127_{5\%}$ NP (right). Scale bar: 100 nm. **c** Peak fluorescence intensity and **d** Young's modulus values of NPs with different hydrodynamic diameters of PLGA core. ($n = 3$). **e** Atomic force microscopy (AFM) images of Lip-$F127_{5\%}$ NPs, $PLGA_{70}$-Lip-$F127_{5\%}$ NPs, and $PLGA_{160}$-Lip-$F127_{5\%}$ NPs and the corresponding deformation images of the NPs after being subjected to forces of different magnitudes. Scale bar: 200 nm

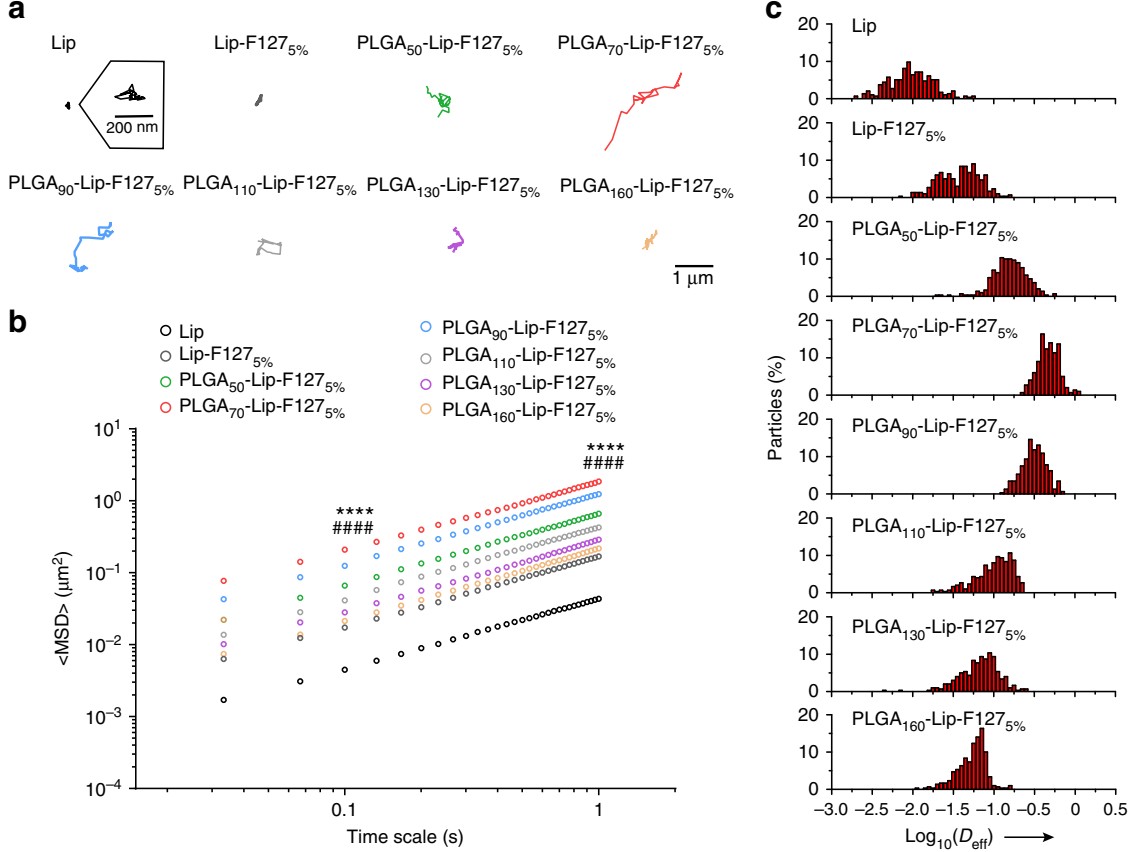

**Fig. 2** Transport of NPs with various rigidities in rat mucus ex vivo. **a** Representative trajectories of particle motion in 1 s in fresh rat intestinal mucus. **b** <MSD> values as a function of time for particles in mucus. **c** Distributions of the logarithms of individual particle effective diffusivity ($D_{eff}$) values in mucus on a time scale of 1 s. The data represent three independent experiments, with each experiment tracking 100 NPs. ****$P < 0.0001$, $PLGA_{70}$-Lip-$F127_{5\%}$ compared to Lip-$F127_{5\%}$, and ####$P < 0.0001$, $PLGA_{70}$-Lip-$F127_{5\%}$ compared to $PLGA_{160}$-Lip-$F127_{5\%}$

investigated the cellular uptake of NPs using human colon carcinoma cell line (Caco-2), in situ pancreatic adenocarcinoma cell (BxPC-3), and human pancreatic star cells (HPSC), which are not covered with biological hydrogels. In this study, the hard NPs displayed the highest cellular internalization (Fig. 3b, Supplementary Note 4 and Supplementary Fig. 4), consistent with theoretical predictions[32].

To visualize the mucus penetration and cellular internalization properties of the NPs, we obtained three-dimensional (3D) fluorescent images of E12 cell monolayers treated with 1,1'-dioctadecyl-3,3,3',3'-tetramethylindocarbocyanine perchlorate (DiI)-labeled NPs (Fig. 3c). Notably, the semi-elastic NPs were found in all layers from the apical to the basolateral side, indicating the effective permeation of these NPs through the mucus and the cell monolayer. In contrast, the hard NPs were seldom observed in deep layers along the z-direction, suggesting decreased cellular internalization because of poor mucus permeation. Additionally, most of the soft NPs were localized in the upper mucus layer on the apical side, and few were observed in the cell monolayer. For 2D visualization, we used 3,3'-dioctadecyloxacarbocyanine perchlorate (DiO; green) and DiI (red) to label the PLGA core and the lipid shell, respectively. After incubation for 2 h, E12 cells incubated with semi-elastic NPs displayed higher red (lipid shell) and green (PLGA core) fluorescence intensities than those incubated with hard NPs (Fig. 3d). Soft NPs incubated with E12 cells showed less red fluorescence than their semi-elastic and hard counterparts. These results imply that in the presence of a mucus layer, the uptake capacity of NPs with different rigidities is as follows: semi-elastic

> hard > soft. Combined together, our results suggest that in E12 cells, the mucus acted more like a barrier to soft and hard NPs, but not to semi-elastic NPs.

We then used BxPC-3 and HPSC multicellular spheroids to evaluate the penetration behavior of NPs in tumors, another common tissue covered with biological hydrogel (Fig. 3e). For the soft and hard NPs, red fluorescence mostly occurred on the periphery of the multicellular spheroids (MCSs) at a scanning depth of 30 μm. In contrast, the penetration capability of the semi-elastic NPs was significantly greater, and red fluorescence inside the MCSs could be clearly observed even at a scanning depth of 70 μm. Therefore, the superior permeability of the semi-elastic NPs was demonstrated to be independent of types of biological hydrogel tested.

**NP entry into tissues covered with hydrogels in vivo.** We next examined whether the exceptionally hydrogel penetration of semi-elastic NPs could facilitate efficient entry into mucosal and tumor tissues in vivo. First, we examined the cross-sectional distribution of NPs using an in vivo intestinal loop incubation method in rats (Fig. 4a, Supplementary Note 5 and Supplementary Fig. 5). The semi-elastic NPs were distributed evenly throughout the intestine and also throughout the mucus layer; consequently, some semi-elastic NPs were observed both in the epithelium and enterocytes, indicating that these NPs might cross the epithelium via enterocytes. However, both the soft and hard NPs remained clumped in the intestinal lumen, and appeared unable to cross the epithelial barrier efficiently. To further confirm the distinct behavior of the semi-elastic NPs, different

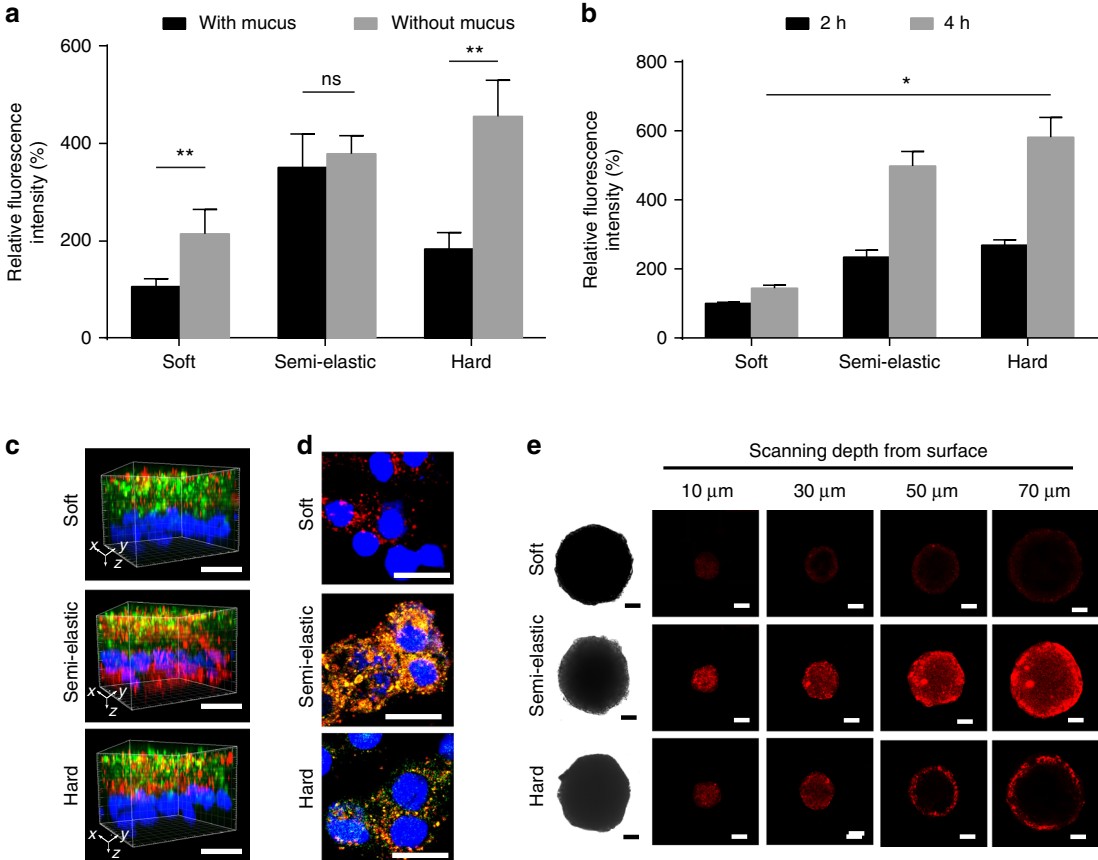

**Fig. 3** Cellular penetration and uptake in vitro. **a** The internalization of NPs by E12 cells with or without pretreatment to remove mucus. Soft: Lip-F127$_{5\%}$ NPs; Semi-elastic: PLGA$_{70}$-Lip-F127$_{5\%}$ NPs; Hard: PLGA$_{160}$-Lip-F127$_{5\%}$ NPs. **b** Caco-2 cell-associated fluorescence after different incubation times. **c** 3D images of the mucus penetration and cellular internalization of NPs in the E12 cell monolayer. Green: mucus stained with Alexa Fluor 488-wheat germ agglutinin. Blue: nuclei stained with Hoechst stain. Red: DiI-labeled NPs. Scale bar: 20 μm. **d** Fluorescent confocal images of E12 cells (stained with 4′,6-diamidino-2-phenylindole [DAPI]) incubated with NPs for 2 h. Colocalization of the PLGA core entrapping DiO (green) and the lipid layer labeled by DiI (red) is indicated in yellow. Scale bar: 20 μm. **e** NP penetration into the BxPC-3 and HPSC multicellular spheroids. Z-stack images were obtained starting from the top and into the core of the spheroid at intervals of 20 μm. Scale bar: 50 μm. Data are shown as the means ± standard deviations (SDs). ($n = 3$) (ns, $P > 0.05$; *$P < 0.05$; **$P < 0.01$)

fluorophore-labeled particles were orally co-administered in rat. The semi-elastic NPs were dispersed widely throughout the intestine, but both the soft and hard NPs remained aggregated in the intestinal lumen. The in vivo results confirmed that the semi-elastic NPs maintained their superiority in terms of mucus penetration and cellular uptake compared with the soft and hard NPs (Supplementary Note 6 and Supplementary Fig. 6).

Figure 4b shows the accumulation behavior of the peritumorally injected NPs. The soft and hard NPs displayed weak fluorescence from the tumor at 6 h post-injection, which were further decayed with time. In sharp contrast, the semi-elastic NPs exhibited a strong fluorescence signal, indicating that semi-elastic NPs could be efficiently internalized into the tumor and stably accumulated within cancer cells. To better understand the rigidity dependence of the intratumoral distribution of NPs, tumors were excised from the mice at 6 h after peritumoral injection and sliced for imaging. The semi-elastic NPs were detected throughout the tumors, whereas other NPs were detected only in the perivascular sections of tumors (Fig. 4c). The superiority of the semi-elastic NPs in tumor penetration and retention was also confirmed via tail vein injection (Supplementary Fig. 7). The cumulative fluorescence intensity of NPs detected increased between 4 and 12 h and showed an upward trend between 12 and 24 h (Supplementary Fig. 7a). At 4 h, Lip showed little fluorescence and was completely metabolized at 24 h. Other samples showed

some fluorescence in tumors at 24 h. The semi-elastic NPs showed the highest tumor accumulation among the NPs studied, and most of the preparations remained in the tumor at 24 h. The soft NPs could not go through the blood vessels, displaying the weakest fluorescence signal in tumors. These in vivo studies confirmed that the rigidity of NPs plays an active role in overcoming both hydrogel and cellular barriers, and in determining the fate of NPs in biological systems.

**In vivo pharmacokinetic study in rats.** Doxorubicin (Dox) is the first-line therapy for a variety of hematologic malignancies and solid tumors. Currently, Dox delivery is limited to intravenous administration because of its extremely low oral bioavailability (approximately 5%)[33]. However, oral Dox delivery is preferred because of the potential increase in patient compliance, improved efficacy, and reduced side effects. To test the possibility of using these NPs as drug delivery vehicles for oral anticancer therapy, Sprague-Dawley rats were orally administered either free Dox or Dox-loaded NPs. The mean plasma concentration time profiles of different Dox formulations are shown in Fig. 5, and the pharmacokinetic parameters are listed in Supplementary Table 4. Notably, the Dox-loaded semi-elastic NPs achieved the highest $C_{max}$ value, which was 14.8-fold, 4.9-fold, and 2.8-fold greater than the values obtained by free Dox, Dox-loaded soft NPs, and Dox-loaded hard NPs, respectively. Similarly, the area under the

**Fig. 4** NP entry into tissues covered with hydrogels in vivo. **a** Distribution of NPs in rat intestines. Intestinal ligated loops were incubated with DiI-labeled NPs for 1 h. Fluorescence micrographs were obtained from transverse cryo-sections of the intestines. "L" indicates the lumen of the intestine. Blue: nuclei of the intestinal villi. Red: NPs. **b** In vivo imaging of BxPC-3 and HPSC tumor-bearing mice taken at the indicated time points before and after peritumoral injection of IR783-labeled soft, semi-elastic, or hard NPs. **c** Distribution of DiI-labeled softer, semi-elastic, and harder NPs in tumor slices of BxPC-3 and HPSC tumor xenografts 6 h after peritumoral injection. The cell nuclei were counterstained with DAPI. The tumor vessels were labeled with anti-CD31 antibody. Scale bars: 50 μm. Softer: Lip-F127$_{5\%}$ NPs; Semi: PLGA$_{70}$-Lip-F127$_{5\%}$ NPs; Harder: PLGA$_{160}$-Lip-F127$_{5\%}$ NPs. ($n = 3$)

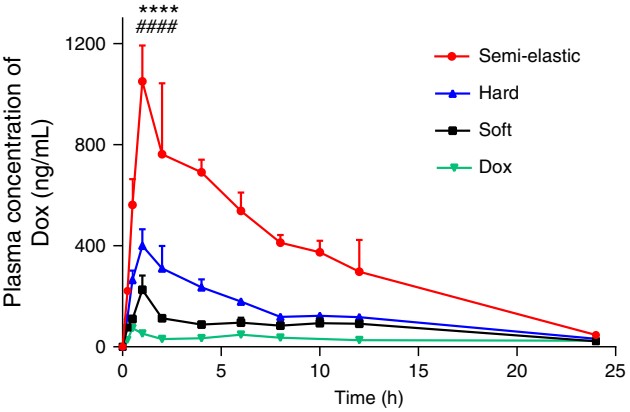

**Fig. 5** Plasma concentration vs. time profile of different Dox formulations after oral administration in a dose equivalent to 10 mg/kg of free Dox. Data are shown as the means ± standard deviations (SDs). ($n = 3$) ****$P < 0.0001$, PLGA$_{70}$-Lip-F127$_{5\%}$ compared to Lip-F127$_{5\%}$, and ####$P < 0.0001$, PLGA$_{70}$-Lip-F127$_{5\%}$ compared to PLGA$_{160}$-Lip-F127$_{5\%}$

curve (AUC) for the semi-elastic NPs was 8.1-fold, 4.0-fold, and 2.9-fold higher than the areas calculated for the free Dox solution, soft NPs, and hard NPs, respectively. Therefore, the superior mucus-penetrating and cellular uptake abilities of the semi-elastic NPs can enhance Dox absorption, and semi-elastic NPs can thus serve as a valuable drug delivery system for drugs with low bioavailability.

**Super-resolution microscopy analysis for NPs**. High-speed confocal imaging by super-resolution microscopy enabled us to observe the 3D mobility and morphology of NPs in hydrogels directly. As shown in the Fig. 6 and Supplementary Movies 1–3, the semi-elastic NPs deformed into ellipsoids and displayed rotational motion in hydrogel, thereby achieving the highest diffusivity. In contrast, the soft NPs deformed excessively and irregularly, causing them to intersperse with the gel fibers and become confined within them; thus, they exhibited the lowest permeation. Finally, the hard NPs almost completely retained their spherical shape during the diffusion process; thus, they displayed moderate diffusivity.

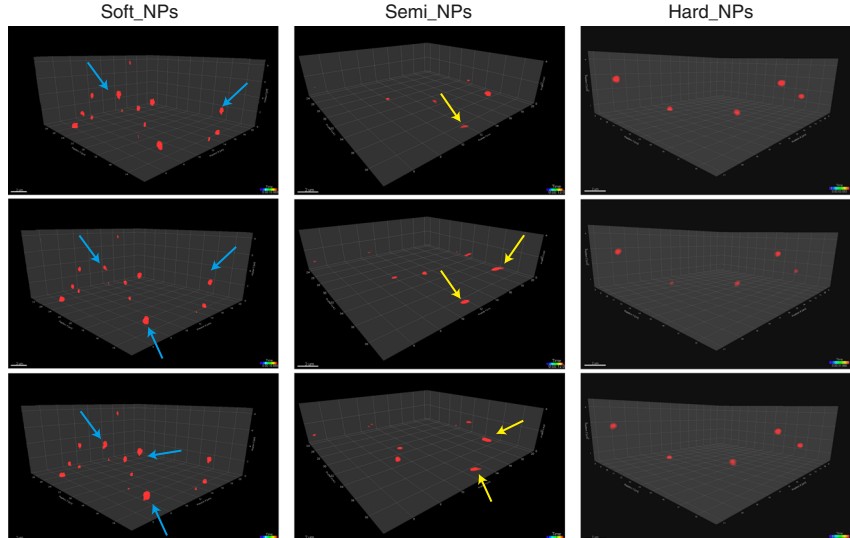

**Fig. 6** Snapshots and trajectories of NPs in rat intestinal mucus, as imaged by Airyscan microscopy. Blue arrows represent soft NPs with irregular shapes. Yellow arrows represent semi-elastic NPs deformed into ellipsoids. The three scans for the soft, semi and hard NPs are similar. Scale bars: 2 μm

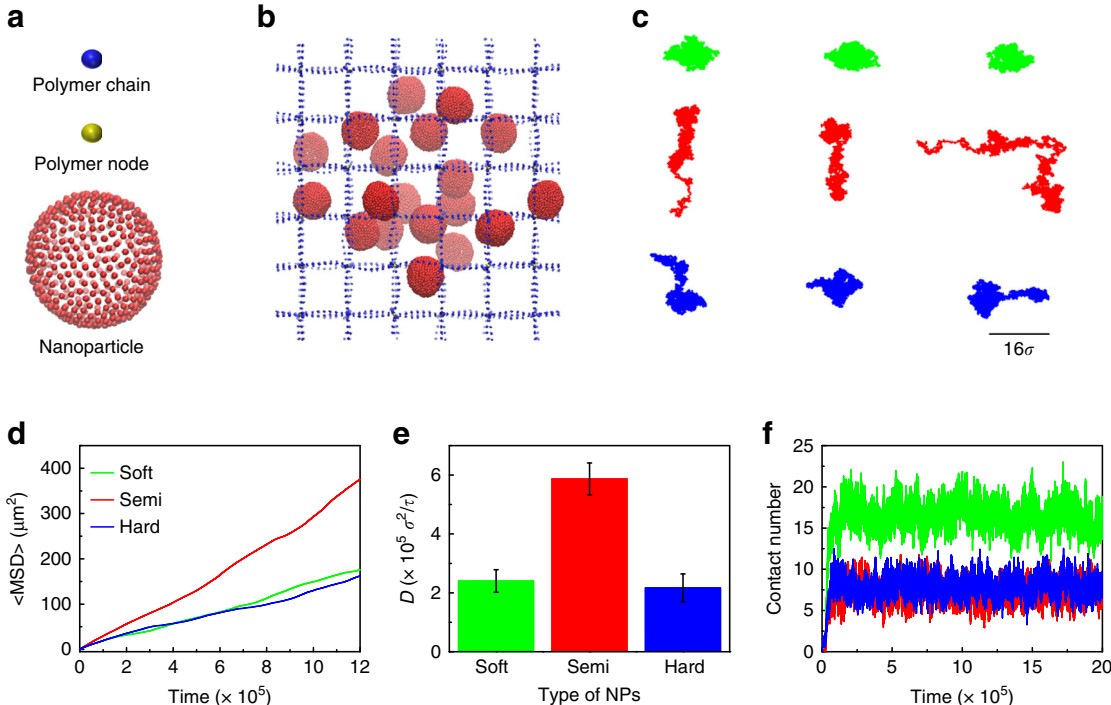

**Fig. 7** Snapshots and trajectories of NPs in mucus fibers. **a** The original construction of NP. **b** A snapshot of the polymer network (blue and yellow) and NPs during the simulation. **c** Typical centroid trajectories of NPs in simulations. **d** The representative MSD values of the three types of NP in a polymer network with a mesh size of $16\sigma$. **e** The diffusivities of NPs in the polymer network. Data are shown as the means ± standard deviations (SDs). ($n = 3$). **f** The number of beads in each NP that came into contact with the polymer network. The blue, red, and green labels represent hard, semi-elastic, and soft NPs, respectively

**MD simulations**. To further elucidate the mechanism underlying the superior gel-penetrating ability of semi-elastic NPs, we conducted coarse-grained (CG) MD simulations. We constructed a model system composed of cross-linked polymers and NPs. A regular polymer network was utilized to represent biological hydrogel fibers with a mesh size of $16\sigma$, where $\sigma$ is the unit of length. The size of the simulation box was $98 \times 98 \times 98\sigma^3$, and periodic boundary conditions were applied in all three directions. Each simulation system involved 108 cross-linked polymer chains

and 27 NPs. The NPs were randomly distributed throughout the polymer network, as shown in Fig. 7b. Three types of NPs with tuned rigidities but the same diameter ($10\sigma$) were studied in our simulations (Fig. 7a, b). The rigidity of the NPs was modified by varying the interaction parameter between each pair of the NPs beads (Supplementary Note 9 and Supplementary Tables 5–6). The simulation was performed using the Large-scale Atomic/Molecular Massively Parallel Simulator (LAMMPS) package[34].

Figure 7b depicts typical snapshots of the diffusion of NPs in the polymer network. Figure 7c indicates the typical translational trajectories for the hard, semi-elastic, and soft NPs. Notably, the semi-elastic NPs diffused faster than the hard and soft NPs. We repeated the simulation 4 times with different starting configurations and obtained the MSD values for semi-elastic, hard, and soft NPs (Fig. 7d), and found that the MSD values of the semi-elastic NPs are higher than those of the soft and hard NPs. The calculated diffusivity of the semi-elastic NPs was approximately 2.5-fold higher than that of the hard and the soft NPs (Fig. 7e).

To explain why the rigidity of NPs can modulate their diffusivity in polymer fibers, we examined the representative structure of NPs. We observed that during transportation, due to their low rigidity, the soft NPs are apt to undergo deformation which could increase the contact area between NPs and hydrogel. As a result, these NPs would suffer from the physical entanglement or increased adhesion, which led to the lowest diffusivity. Indeed, examining the number of contacts between the beads of one NP and the polymers (Fig. 7f) revealed that the soft NPs had approximately 17 contacts with the polymers, which was double the number of contacts found for the hard NPs and for the semi-elastic NPs. Therefore, we concluded that the excessive deformation of soft NPs induces a high affinity for the polymers, thereby restricting their diffusion ability. In contrast, semi-elastic NPs tended to deform into ellipsoids because of their moderate stiffness, and this deformation did not induce excessive contact between the NPs and the polymer (Fig. 7f). We have recently applied molecular simulations and stimulated emission of depletion (STED) microscopy to reveal that the movement of ellipsoidal NPs in a mesh structure consists of two parts, rotation around the polymers and translational diffusion, which contributes to their superior diffusivity over spherical NPs[35]. The elongated shape enabled the ellipsoidal NPs to avoid becoming trapped by the polymer and therefore achieve higher diffusivity than their spherical counterparts. To elucidate how the shape change of NPs influenced their diffusion, we also simulated the diffusion of rigid ellipsoids with the same volume as the sphere NPs in the same polymer network. The results showed that the diffusivity of the rigid NPs (one spherical NP and two ellipsoidal NPs) is influenced by their aspect ratio (AR) (Supplementary Note 13 and Supplementary Fig. 14). The NPs diffuse faster as their AR increases. For a spherical semi-elastic NP, the diffusivity is smaller than that of a rigid ellipsoidal NP with AR = 3, and close to that of a rigid ellipsoidal NP with AR = 2 (Supplementary Fig. 14). These results further suggested that the semi-elastic NPs could enhance diffusivity by changing their shape from spheres to ellipsoids during diffusion. This mechanism explained the anomalous phenomenon observed in the experiments and simulations in which semi-elastic NPs exhibited greater permeation than both soft and hard NPs.

To further verify that the hydrogel penetration could be exclusively enhanced by modulating the rigidity of NPs, we compared the movement of core-shell polymer-lipid NPs with liposomes of different sizes or mucus-penetrating particles (MPPs) with identical hydrodynamic sizes (Supplementary Note 7–8, Supplementary Figs. 8–9, and Supplementary Table-2–3). Decreasing the size of NPs is an effective strategy for enhancing their diffusivity in hydrogel[36,37]. We found that the MSD values of the NPs increased slightly as the particle size decreased. However, after encapsulating the PLGA cores into liposomes, the MSD values of the semi-elastic NPs were dramatically increased (approximately by 100-fold). Dense PEG coatings have been demonstrated to facilitate the rapid penetration of NPs into hydrogel by minimizing adhesive interactions between NPs and hydrogel constituents[38]. The three types of MPPs (i.e., PEGylated liposomes[39], PEGylated PLGA NPs[40]

(PLGA-PEG$_{5\%}$), and PLGA$_{70}$-Lip-F127$_{5\%}$ NPs) displayed excellent penetration capabilities in rat intestinal mucus; however, the semi-elastic NPs performed best in terms of their diffusion behavior. Altogether, these indicate that the rigidity of NPs plays a key role in their transport across mucus.

## Discussion

By exploring how the NPs overcame the multiple barriers, we gained insight into the gate-keeping functions of hydrogels and the cell. We found that the critical barrier to mucosal and tumor delivery could be overcome by modulating the rigidity of NPs. For soft NPs, biological hydrogels and the cell act as two separate yet equally critical barriers; indeed, these NPs showed both weak hydrogel penetration and low cellular uptake. Furthermore, hard NPs could not penetrate hydrogels efficiently, although they showed the best cellular uptake. Thus, hydrogels are the main barriers to the penetration of hard NPs. Our in vivo experiments demonstrated that oral delivery of semi-elastic NPs were able to efficiently deliver Dox, suggesting that the strategy of modifying the rigidity of NPs to overcome the multiple barriers of tissues and organs holds promise for drug delivery. In addition, adding a solid core with suitable size to (soft) liposomes would be helpful on many levels, from diffusion to uptake, and further for drug delivery especially for overcoming the multi-biological barriers. We anticipate that these results will inspire the rational design of novel NP-based drug delivery systems in the near future.

## Methods

**Experimental reagents**. PLGA copolymers (LA/GA molar ratio 75:25; Mw ~60 kDa) were purchased from Advanced Polymer Materials Inc. (Canada). PLGA$_{45k}$-PEG$_{5k}$ and PLGA$_{15k}$-PEG$_{5k}$ were purchased from Daigang Biomaterial Co., Ltd. (China). Egg phosphatidylcholine was purchased from Q.P. Corp (Japan). Cholesterol was purchased from Sinopharm Chemical Reagent Co., Ltd. (China). Pluronic F127 was received as a gift from BASF (Germany). DSPC and 1,2-distearoyl-sn-glycero-3-phosphoethanolamine-N-methoxy(PEG)-2000 (DSPE-PEG2000) were purchased from Avanti (USA). DAPI and DiI were purchased from Beyotime BioTechnology (China). Alexa Fluor 488-conjugated WGA was obtained from Sigma Aldrich (USA). HT29-MTX-E12 (E12) cells (52nd–56th passages) cultured for 14–18 days were supplied by the ADME Department of Novo Nordisk A/S. Caco-2, BxPC-3, and HPSC cell lines were obtained from the American Type culture collection (ATCC, Manassas, VA, USA).

**Cell culture**. HT29-MTX-E12 (E12) cells were maintained in high-glucose Dulbecco's modified Eagle's medium (DMEM; Gibco) with 10% (v/v) fetal bovine serum (FBS; HyClone), 1% (v/v) nonessential amino acids, 1% (v/v) L-glutamine, and 1% penicillin and streptomycin (100 IU/mL) at 37 °C in 5% CO$_2$. To study the uptake of NPs quantitatively, E12 cells were seeded into 24-well plates (50,000 cells/well) and cultured in growth medium at 37 °C in a 5% CO$_2$ incubator until they adhered to the plates. To establish cell monolayers for qualitative studies, the E12 cells were seeded at a density of $10 \times 10^4$ cells/mL on a polycarbonate membrane (pore size: 0.4 µm; diameter: 6.5 mm; cell growth area: 0.33 cm$^2$) in Costar Transwell 24-well plates (Corning Costar Corp.). Transepithelial electrical resistance (TEER) was monitored using an electrical resistance meter (Millicell ERS-2, Millipore) to ensure the integrity of the cell monolayers.

**BxPC-3 and HPSC multicellular spheroids**. The BxPC-3 and HPSC multicellular spheroids were cultured using a lipid overlay system as described previously[41]. Briefly, agarose solution was prepared in sterile water (2% w/v) and sterilized in an autoclave. Each well of 48-cell culture plates was coated with a thin layer (200 µL) of sterilized agarose solution and seeded with 5000 BxPC-3 cells and 5000 HPSC cells in 500 µL DMEM medium containing 1% penicillin−streptomycin as the culture media. Then, the plates were gently agitated for 2 min, and BxPC-3 and HPSC multicellular spheroids were allowed to grow for 7 days at 37 °C in the presence of 5% CO$_2$.

**Animal care**. Male Sprague–Dawley rats (200–250 g) were provided by the Animal Experiments Center of the Shanghai Institute of Materia Medica (Shanghai, China). Animals had free access to rat chow and tap water. All animal experiments were conducted according to the Institutional Animal Care and Use Committee (IACUC) guidelines of the Shanghai Institute of Materia Medica (IACUC code, 2015-12-GY-21).

**Fabrication and characterization of NPs.** PLGA cores with variable hydrodynamic diameters (50, 70, 90, 110, 130, and 160 nm) were first fabricated using a microfluidic-based system (Nanoassemblr™). PLGA was dissolved in acetone to prepare a PLGA solution (5 mg/mL), and Pluronic F68 was dissolved in deionized water to form an aqueous solution (5 mg/mL). To fabricate PLGA NPs, we employed syringe pumps to introduce the PLGA solution into the left inlet and the aqueous solution into the right inlet of the microfluidic mixer (Nanoassemblr™). To prepare PLGA NPs with different hydrodynamic diameters, we simply changed the total flow rate (2 mL/min to 15 mL/min to produce NPs with hydrodynamic diameter from 50 nm to 160 nm). The bare PLGA cores were subsequently encapsulated into liposomes using a microextruder device (LiposoFast LF1) to obtain the final core-shell PLGA-lipid NPs. Phosphatidylcholine, cholesterol and Pluronic F127 (1/28/5, molar ratio) were dissolved in 5 mL of chloroform. The solution was then transferred to a 100-mL round flask, and the chloroform was removed by a rotary evaporator at 25 °C in a vacuum to form a thin lipid film. Trace chloroform was cleared by flushing with nitrogen for 1 min. The dry lipid film was then hydrated at 25 °C for 1 h by adding 20 mL of distilled water or PLGA NP suspension to obtain crude liposome suspensions. The crude liposome suspensions were finally extruded through a 200-nm polycarbonate membrane using a microextruder device to obtain the PLGA-lipid NPs.

**Cryo-TEM.** Cryo-TEM was performed using a Tecnai 12 electron microscope to differentiate the structures of the NPs. A drop of NP suspension was placed onto a carbon-coated copper grid and blotted. The samples were then shock-frozen by rapid immersion in liquid ethane, and they were cooled to 90 K by liquid nitrogen. The specimens were transferred to the Tecnai 12 electron microscope and analyzed at 120 kV.

**AFM.** AFM imaging and force measurements were conducted using a Bio-FastScan scanning probe microscope (Bruker) in PeakForce QNM imaging mode. The NP suspensions were dropped onto a freshly cleaved mica surface and air-dried at room temperature in 85% humidity for 1 h. The samples were imaged at a scan rate of 1 Hz. A cantilever with a deflection sensitivity of 99 nm/V and a tip with a spring constant of 2 N/m were applied. All images of NPs obtained from AFM were processed by the software NanoScope Analysis (Bruker), and the Young's modulus was calculated by processing the images of NPs without deformation in PeakForce QNM mode ($n = 3$).

**Water content studies.** To label the water inside the NPs with core-shell structures, we applied a 200 μm ethyl rhodamine B solution or NP suspension as the hydration fluid. After synthesis, free ethyl rhodamine B in solution was removed from the NPs using a Sephadex G50 size-exclusion column, and the collected fractions were diluted to the same concentration for fluorescence intensity analysis. The fluorescence emission spectra of the separated NPs were measured at an excitation wavelength of 510 nm using a synergy microplate reader.

**Multi-particle tracking.** Ex vivo tracking experiments were conducted as previously described[42]. For mucus collection, the small intestine was excised after euthanizing the rats, and approximately 1.5–2 mL of mucus from each fasted rat was collected. The properties of intestinal mucus collected from rodents have been reported in our previous work[35]. The average size of the mucus pore was 186 nm, and the majority of pores were more 100–240 nm in diameter. In this study, ten rats were sacrificed to collect mucus for all particle tracking studies. A volume of 5 μL of NPs (100 μg/mL) was pipetted on the mucus or PEO hydrogel (200 μL), and incubated for 0.5 h prior to microscopy. Movies were captured at a temporal resolution of 32.6 ms for 10 s using an inverted fluorescence microscope (DMI 4000B, Leica, Germany). The tracking resolution was approximately 10 nm, determined by gluing polystyrene (PS) microspheres on microslides and tracking their apparent displacement, as previously reported[43]. Fifteen movies were taken and analyzed for each NP. The trajectories for $n = 100$ particles were analyzed using ImageJ for each experiment, and three independent experiments were performed. Particle-averaged mean squared displacement (MSD) and effective diffusivities ($D_{eff}$) were calculated using the following equations:

$$\mathrm{MSD}_t = \frac{\sum\left[(x_t - x_0)^2 + (y_t - y_0)^2\right]}{n} \qquad (1)$$

$$D_{eff} = \frac{\sum[\mathrm{MSD}_t/(4t)]}{n} \qquad (2)$$

(where $x$ and $y$ represent the coordinates of the particle, and $t$ = time scale or time lag).

**Ex vivo mucus-penetrating property measurement.** For 2D mucus coverage and 3D mucus-penetrating observations, the small intestine extracted from male Sprague−Dawley rats was cut into small loops (2.5 cm in length) and ligated by medical suture. After staining the mucus with Alexa Fluor 488 conjugated-WGA, 100 μL of PBS solution containing DiI-labeled NPs (100 μg/mL) was slowly injected, and the intestinal loops were incubated in Krebs-Henseleit (KH) buffer for 30 min at 37 °C. The sliced intestinal sections were inspected via CLSM. The mucus coverage area of red fluorescence was measured using the ImageJ software.

**Nanoparticle penetration in MCSs.** MCSs were transferred to ultralow-attachment 24-well plates and incubated in DMEM medium at pH 7.0. The fluorescence-labeled Lip-F127$_{5\%}$ (softer), PLGA$_{70}$-Lip-F127$_{5\%}$ (semi-elastic), PLGA$_{160}$-Lip-F127$_{5\%}$ (harder), Lip-80 and Lip-200 NPs were added to the medium with the DiI concentration adjusted to 1 ng/mL. After 2 h of incubation, the spheroids were harvested and washed with PBS (pH 7.4, 0.01 M) three times and then observed by confocal laser scanning microscopy (CLSM).

**Cellular uptake.** To quantify cellular uptake and investigate the influence of the mucus layer, E12 cells were separated into two groups: one group was pretreated with NAC to remove the mucus layer, and the other group was washed gently with phosphate-buffered saline (PBS) to preserve the mucus layer as much as possible. Both cell groups were incubated for 7 days and then treated with DiI-labeled NPs (250 μg/mL) for 2 h. Then, cell lysis buffer was added, and cell samples were collected for further analysis by flow cytometry or by using a bicinchoninic acid (BCA) assay kit.

To investigate cellular uptake qualitatively, cells were incubated with DiI-labeled NPs (250 μg/mL). After 1 h incubation, we stained the nuclei with Hoechst stain and the mucus with Alexa Fluor 488-conjugated WGA. Subsequently, the cells were washed twice with Hank's balanced salt solution (HBSS) and observed by confocal laser scanning microscopy (CLSM).

**Nanoparticle distribution in rat small intestine.** To investigate intestinal distribution, male Sprague–Dawley rats were anesthetized with urethane solution, and the ileum was exposed from a small incision in the abdomen. A 2-cm region was tied off using surgical sutures, and 200 μL of NP suspensions was injected into the loop using a syringe. The loops were excised 1 h after administration of NPs and frozen in optimum cutting temperature (OCT), then sliced at a depth of 20 μm. The tissue slices were briefly fixed in 4% paraformaldehyde, stained with DAPI, and examined via CLSM.

**Tumor penetration study in vivo.** A subcutaneous-tumor-bearing nude mouse model was established by inoculating $3 \times 10^6$ BxPC-3 and $3 \times 10^6$ HPaSteC cells in 100 μL of PBS into the subcutaneous tissue of the right hind limbs of nude mice. When the tumors were ready for use, three groups of nude mice were subject to peritumoral injection with 100 μL of IR783-labeled Lip-F127$_{5\%}$, PLGA$_{70}$-Lip-F127$_{5\%}$, and PLGA$_{160}$-Lip- -F127$_{5\%}$ NPs. In vivo radioactive optical imaging was performed with an IVIS 100 Spectrum system (PerkinElmer, USA). The mouse was placed in a light-tight chamber under isoflurane anesthesia, and luminescent images were acquired over 5 min at 1–24 h postinjection. Images were analyzed using Living Image 3.1 software.

For the tumor penetration analysis, mice bearing BxPC-3 and HPSC xenografts were peritumorally injected with DiI-labeled Lip-F127$_{5\%}$, PLGA$_{70}$-Lip-F127$_{5\%}$, and PLGA$_{160}$-Lip-F127$_{5\%}$ NPs (1 mg/mL). Six hours later, the mice were sacrificed, and tumors were excised and washed with precooled saline and the surface water was extracted with the filter paper. Excised tumor were frozen in OCT medium (Sakura Finetek, USA) at −80 °C. The tumor tissue was cut into 10 μm thick slices (Leica CM1950, Germany). Finally, the slides were fixed with 4% paraformaldehyde, cell nuclei were counterstained with DAPI, and immunofluorescence staining with anti-CD31 antibody was utilized to label tumor vessels. Fluorescence signals were imaged under a CLSM (FV1000, Olympus, Japan) to investigate the tumor penetration of various NPs.

**In vivo pharmacokinetic study.** Male Sprague–Dawley rats (200–220 g) were fasted overnight with free access to water and were randomly distributed into four groups ($n = 3$ for each studied group). Adriamycin and Dox-loaded NPs at a drug dose of 10 mg/kg body weight were administered to the rats by oral gavage. Blood samples (approximately 0.5 mL) were collected from the retro-orbital plexus at 0.25, 0.5, 1, 2, 4, 6, 8, 10, 12, and 24 h in microcentrifuge tubes containing heparin. These blood samples were centrifuged at 3500 r.p.m. for 10 min to separate the plasma, and the plasma samples were modified for high-performance liquid chromatography (HPLC) to measure Dox levels.

**Observation of deformation of NPs by LSM 880 with Fast Airyscan in 3D.** To observe the 3D deformation of NPs during their penetration through mucus, images, and movies were acquired by super-resolution microscopy with 40 z-stack slices and at 30 time points using a Fast Airyscan confocal microscope (ZEISS LSM 880 with Fast Airyscan, ZEISS, Germany) equipped with a Plan-Apochromat 63X/1.4 numerical aperture oil objective in Fast Airyscan mode. All movies were captured using ZEN software (ZEISS).

**MD simulation.** In this work a regular cross-linked polymer network mesh size of $16\sigma$ was constructed to represent mucin fibers. Each fiber was composed of a series of beads spanning the entire simulation box. Different fibers were cross-linked by a

node bead to simulate entanglement and crosslink of mucin fibers. The bonded interaction between neighboring beads along the polymer network is described by a harmonic spring force. The NP with the size of $10\sigma$ is modeled using the one-particle-thick model. As usual, the units of length, mass, time, and energy are the bead radius $\sigma$, bead mass $m$, $\tau$, and $\varepsilon$ respectively.

In all simulations, the Lennard–Jones (LJ) potential was used to describe the non-bonded interactions between two beads except the interaction of each pair of NPs' beads. Following the notation from the original paper, the inter-particle interaction between the NPs' beads is described by a soft-core pairwise potential with the interaction strength weighed by the relative orientations of the particle pair[44]. The details of the setup of the system and the interaction potential were listed in Supplementary Note 9–10. The interaction parameter between NPs' beads and polymer chains was verified by a series of full-atom molecular dynamics simulations, with the model and method given in Supplementary Information. To verify the robustness of our simulation model, we tuned the interaction parameter between the polymer network and NPs from $\varepsilon_{ij}=0.02$ to $\varepsilon_{ij}=0.2$ and the pore size of the network from $14\sigma$ to $18\sigma$ in the simulations; the results were summarized in Supplementary Notes 11–12 and Supplementary Figs. 12–13.

During the simulations, we defined contact between an NP and the polymer network as the condition in which the distance between a NP bead and a polymer network bead was smaller than $3\sigma$. The number of beads contacted by the NPs was then calculated. The MSD values and effective diffusivities were calculated using the following equations:

$$\text{MSD}(t)=\langle(x_t-x_0)^2+(y_t-y_0)^2+(z_t-z_0)^2\rangle \tag{3}$$

$$D_{\text{eff}}=\frac{\text{MSD}(t)}{6t} \tag{4}$$

where $x$, $y$, and $z$ represent the center of mass of a particle, $t$ is the duration of the time lag, and $\langle\cdots\rangle$ represents the average of the NPs. In this work, we repeated each simulation 3 times with different initial configurations and obtained the mean MSD of different type of NPs. In the simulations, we constrained the node bead by applying a spring to tether them to their initial positions. The Velocity-Verlet algorithm was utilized to perform time integration during the simulations. The integration time step was $\Delta t=0.01\tau$. In order to remain intact of the NPs in the simulation, Langevin thermostat was used to control the system temperature at $k_{\text{B}}T=0.23\varepsilon$, where $k_{\text{B}}$ is the Boltzmann constant, and $T$ is temperature. We performed constant number-volume-energy (NVE) integration to update the position and velocity of the beads in each simulation. The total simulation time was $2\times10^6\tau$. After about $8\times10^5\tau$, the MSD calculations were performed. The fitting of the diffusivity $D$ was calculated by linearly fitting the MSD versus time lag from $1.3\times10^6\tau$ to $1.8\times10^6\tau$. The slope of the fitting line is denoted by $k$; thus, $D=k/6$.

**Statistical analysis**. All data are reported as mean ± standard deviation (s.d.). Inter-group differences were analyzed using the Student's $t$-test when two groups were compared, or one-way analysis of variance (ANOVA) with Tukey post hoc test when multiple groups were compared. (ns, $P>0.05$; *$P<0.05$; **$P<0.01$; ***$P<0.001$)

**Data availability**. The data that support the findings reported herein have been deposited in Figshare, https://figshare.com/s/55031aaf880581942163. (DOI: 10.6084/m9.figshare.6152189). Data are also available on reasonable request from the corresponding authors.

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

## Acknowledgements

We are grateful for the financial support from the National Natural Science Foundation of China (81373356, 81573378, and 81773651 to Y.G., 11422215, 11272327, and 11672079 to X.S.), the Strategic Priority Research Program of Chinese Academy of Sciences (XDA01020304 to Y.G.), and the National Science Foundation under Grant CMMI-1562904 (H.G.). This work was also partly supported by K.C. Wong Education Foundation, CASIMM0120153020, New Star Program, Shanghai Institute of Materia Medica, CAS, and the Opening Fund of State Key Laboratory of Nonlinear Mechanics. The computation experiment was mainly supported by the Supercomputing Center of Chinese Academy of Sciences (SCCAS). We thank the National Center for Protein Science Shanghai for our Cryo-EM, and we are grateful to Danyang Li for her help in preparing the EM samples. We also thank the ZEISS Microscopy Lab Shanghai for our light microscope data, and we are grateful to ZEISS China for their help in obtaining data from the ZEISS LSM 880 with Fast Airyscan. The AFM experiments were supported by Huiqin Li from the Instrumental Analysis Center of Shanghai Jiao Tong University (SJTU).

## Author contributions

Y.G., X.S., H.G., M.Y., F.T., and L.X. planned and executed the experiments, analyzed the data, and were involved in discussions of the data. Y.G., X.S., M.Y., F.T., L.X., N.Z., and H.G. wrote the manuscript. L.X. prepared the NPs and performed the multi-particle tracking experiments. M.Y. designed and performed the mucus penetration experiments. F.T., Q.S., and J.W. performed and analyzed the MD simulations. N.Z., A.W, C.Z., X.Z., and S.G. contributed materials. All authors critically reviewed and approved the manuscript.

## Additional information

**Competing interests:** The authors declare no competing interests.

