## [Peer Review File · Nature Communications]

Reviewers' Comments:

Reviewer #1:

Remarks to the Author:

This paper certainly presents important results and provides an explanation for those results by exploring a simple simulation model. Since I am not familiar with most of the literature related to nanoparticle transport through mucus, I cannot say if it is original or not. Assuming that it is sufficiently original, I recommend it for publication with a few changes.

Figure 6 g (line 323) does not really belong in the same plot as the simulation results. This confused me when I read the manuscript. I think it would be better to place it in a separate figure. In addition, I think that the paragraph on experimental imaging of the nanoparticles (lines 351 to 361) should be placed before the paragraph on MD simulations and appropriate modifications to those paragraphs made. The MD simulations should probably be presented as supporting the experimental results, not the other way.

Given that the effectiveness of the semi-elastic particles seems to be related to their ability to deform into ellipsoidal shapes, there is a question of whether shape or elasticity is more important. Are rigid ellipsoidal particles as effective as spherical semi-elastic particles? Although it may not be feasible to construct real ellipsoidal particles of the same type found in this paper, rigid ellipsoidal particles could be simulated. If the authors do not choose to simulate rigid ellipsoidal particles, they should at least mention the possibility that they might also be effective or why that is unlikely and provide supporting references.

Line 467 mentions "Time-averaged mean squared displacement". It should be "Particle-averaged mean squared displacement".

The equations for MSD at time t and diffusivity (lines 469 and 553) should indicate somehow that an average is taken over all of the particles.

For fitting the simulation MSDs, it is stated that the MSDs were fit starting from time 0 (line 563). Generally, there is a short time over which the MSD is not expected to be linear. Therefore, fitting should usually not start from time 0. This nonlinearity at short times can be seen in Figure 6 d. Given the length of the trajectories, throwing away this short time will not have a large effect on the results. However, one way to determine this time is to plot the derivative of the log of MSD with respect to the log of time. Once that begins fluctuating around a value of 1, then the MSD has become linear. See the attached plot, where the nonlinear part lasts between 10 and 15 ps and the fitting should not be started until after that.

Reviewer #2:

Remarks to the Author:

In this paper, the authors tested their hypothesis that particle diffusion in biological hydrogels would be dependent upon particle rigidity. In particular, they formulated core-shell PLGA-lipid NPs possessing a range of rigidity, as determined by Young's moduli, and demonstrated that semi-elastic NPs were most efficient at mucus and tumor penetration. This paper is relevant to the readership of the journal and easy to read, and experiments are generally well designed and executed. However, the connection between their experimental observation and data interpretation is relatively loose and there are a few critical issues to be addressed, as listed below, prior to its consideration for publication.

Specific comments:

1. The authors interpreted their experimental observation largely based on MD simulation, but the connection does not seem tight. Specifically, there are many assumptions in the simulation without

clear justifications and/or relevant references. For example, based on what rationale the authors set the relative dimensions of particles and mesh pore sizes in the simulation to be 10:16? How the degree of polymer affinity was selected to model biological hydrogels (i.e. mucus vs. tumor, etc.)? Overall, the authors must provide clear rationales for the use of different numbers, equations and theories with appropriate references.

2. While the diffusivity values for soft and hard NPs are comparable in rat intestinal mucus (Figure 2b), MD simulation suggests that hard NPs move much faster than soft NPs (although statistical analysis is not provided) (Figure 6). This discrepancy again raise a concern for using the simulation to interpret the authors' experimental data.

3. The authors claimed that soft NPs exhibited lowest diffusivity as they "deformed excessively and attached themselves to the polymer", which indicates that particle diffusion is largely hindered by adhesive interactions rather than flexibility itself. However, it has been previously reported that NPs can effectively resist adhesion to mucus when their surfaces are densely coated with F127. This raises a question whether their NPs with different rigidities possess equally well shielded surfaces; while z-potential measurement is one way to characterize particle surfaces, exhibiting similar z-potentials among different NP population does not ensure similar/identical NP surface properties. It is conceivable that NPs may efficiently move in a biological hydrogel regardless of rigidity, if their surfaces are well shielded unless NPs are extremely elongated to intertwine with gel fibers. In addition, what would be the role of deformability of NPs on their viscous drag?

4. It is unclear how the rotational behaviors of semi-elastic NPs facilitate their movements in a biological hydrogel. The authors must provide more detailed and precise interpretation of their observations beyond speculation. In addition, the range of Young's moduli tested in this study is relative narrow (i.e. 5 – 110 MPa) and the authors are claiming that doubling the value (i.e. semi-elastic to hard) significantly impacts the NP mobility in hydrogels. However, multiple independent groups have shown that NPs possessing over an order of magnitude greater Young's moduli, such as polystyrene beads, efficiently penetrate mucus and tumor tissues when those particles are small enough and capable of resisting adhesion to the mucus.

5. The author claimed that their NPs with moderate rigidity showed unprecedented diffusivity through mucus compared to any known synthetic NPs, which is an overstatement. The authors formulated some possible MPP formulations and showed that their semi-elastic NPs exhibited greater diffusivity values, but it is unclear whether those control particles were carefully made to efficiently penetrate mucus. For example, z-potentials of PLGA-PEG NPs are markedly more negative (-15 mV) compared to those of core-shell NPs (-5 mV), suggesting that PLGA-PEG are possibly not well shielded by PEG. In addition, it is unclear whether the differences among different formulations are statistically significant.

6. Several necessary details are missing throughout the method section, which makes it hard to precisely gauge the validity of this study and to reproduce the observations here by an independent group. For example, why two different pluronics (i.e. F68 and F127) were used for the formulation of core-shell NPs? how were the Young's moduli of NPs calculated? What was the properties of intestinal mucus collected from rodents (e.g. mucin concentration, etc.) and how many animals were sacrificed to collect mucus? What was the concentration of NPs added to mucus for particle tracking studies? How many movies were taken and analyzed? How the MPP-control particles were made; did the authors strictly follow published methods of formulation and characterization and/or collaborate with relevant groups? How the particle tracking data was processed/averaged? Was the presence of mucus confirmed in the cell culture studies? Were qualitative image-based studies repeated to corroborate the authors' findings or could the authors conduct image-based quantification?

7. The authors used student's t-test for statistical analyses, but multiple comparisons should be made with ANOVA. In addition, statistical analysis is missing throughout the manuscript, particularly for particle tracking studies.

8. In Figure 4, it is unclear whether semi-elastic NPs indeed provided improved distribution in mucus and tumor tissue. While the overall fluorescence intensity was greater in the intestinal lumen for semi-elastic NPs, all NPs seemed to distribute near the epithelial surfaces, indicating efficient mucus penetration. Likewise, while more semi-elastic NPs were accumulated in tumor compared to other NPs following systemic administration, it may be due to the difference in their

blood pharmacokinetics rather than their abilities to penetrate tumor tissue. Comparing the NP distribution following local administration would be a more relevant experiment to demonstrate superior ability of semi-elastic NPs to diffuse in tumor tissues.

9. Why soft NPs retained much shorter in tumor in vivo compared to other NPs if they readily adhere to the tumor tissue as suggested by the authors (Figure 6b)?

10. Rhodamine B does not "label" the water, but visualize the volume occupied by water.

Reviewer #3:

Remarks to the Author:

In their paper "Rapid Transport of Deformation-Tuned Nanoparticles across Biological Hydrogels and Cellular Barriers", the authors show that particles of different rigidities but identical surface chemistry and size can have very different diffusion properties within biological hydrogels. The experiments are straightforward and convincing, notwithstanding some quibbles with Fig. S4 described below. Overall, it is my recommendation that the manuscript be accepted for publication with minor revisions.

More specifically, to me it seems like the results of this paper suggest that a nanoparticle with a 70nm solid core and 200nm shell diffuses essentially as though it were 70nm solid nanoparticle, because it can deform to slip through a 70nm pore by presenting a 70nm cross section. Or more generally, a PLGAx-Lip-F1275% particle could diffuse essentially like a solid nanoparticle of diameter x , except that for small PLGA cores this trend stops holding because the liposome can deform around gel polymers. If this is true, then the natural comparison looking at nanoparticle size changes in Fig. S4 would be to compare PLGA70-Lip-F1275% to hard, rather than soft, smaller nanoparticles. As it is, I am not entirely sure what the motivation for the Fig. S4 experiment is.

At any rate, the discussion may benefit from clarifying the advantages of semi-elastic nanoparticles in potential applications. Compared to hard nanoparticles, this paper shows that semi-elastic particles have a lower effective diameter for diffusion (which is good!), but also have a lower uptake (which is bad!) possibly also essentially due to a smaller effective diameter, so semi-elastic particles are basically equivalent to smaller hard particles. Is the idea just that you can fit more stuff in the semi-elastic particles, which can be bigger than the equivalent smaller, solid particle? That's interesting! And could perhaps be stated explicitly. But overall, the benefits of semi-elastic larger nanoparticles over smaller, hard nanoparticles seem somewhat marginal. However, it appears that adding a solid core to (soft) liposomes would be helpful on many levels, from diffusion to uptake. That's very interesting. So it seems as though the utility of this paper will come largely from applications where a liposome, rather than a solid nanoparticle, is desired for the biological effect. If this is true it is potentially worth stating directly.

Some minor comments:

- I think the capitalization on the title is off, the copy editor can presumably confirm this
- Line 672: it's the Hanes group, not the Justin group
- Line 175: I think the word "frequency" should be deleted here: "Distributions of the logarithms of individual..." seems fine.
- Line 366: to stay consistent it should be "Fig. S5" (the period was left out)
- Figures S4 and S5 are mentioned after Fig. S6-S8, so they should be reordered.
- Figure S9 is never referenced in the main text

Response to Review

We are grateful to the Editorial Office and reviewers for their constructive comments and suggestions. We have carefully revised the paper according to the reviews' comments. The changes are highlighted in the revised manuscript. The following are our detailed responses:

Reviewer #1 (Remarks to the Author):

This paper certainly presents important results and provides an explanation for those results by exploring a simple simulation model. Since I am not familiar with most of the literature related to nanoparticle transport through mucus, I cannot say if it is original or not. Assuming that it is sufficiently original, I recommend it for publication with a few changes.

Response: Thank you for your positive assessment of our work.

Figure 6 g (line 323) does not really belong in the same plot as the simulation results. This confused me when I read the manuscript. I think it would be better to place it in a separate figure. In addition, I think that the paragraph on experimental imaging of the nanoparticles (lines 351 to 361) should be placed before the paragraph on MD simulations and appropriate modifications to those paragraphs made. The MD simulations should probably be presented as supporting the experimental results, not the other way.

Response: Thank the reviewer for making these insightful suggestions, which we have adopted. Figure 6g has been placed in a separate Figure 6 in the revised manuscript. The MD simulations are conducted to reveal the mechanisms that underlie the experimental findings, so we have adopted the reviewer's suggestion, and the paragraph on experimental imaging of the nanoparticles has been placed before the paragraph on MD simulations in the revised manuscript.

Given that the effectiveness of the semi-elastic particles seems to be related to their ability to deform into ellipsoidal shapes, there is a question of whether shape or elasticity is more important. Are rigid ellipsoidal particles as effective as spherical semi-elastic particles? Although it may not be feasible to construct real ellipsoidal particles of the same type found in this paper, rigid ellipsoidal particles could be simulated. If the authors do not choose to simulate rigid ellipsoidal particles, they should at least mention the possibility that they might also be effective or why that is unlikely and provide supporting references.

Response: Thank the reviewer for this constructive suggestion. We have adopted reviewer's suggestion and constructed new simulation model to simulate the diffusion of rigid ellipsoidal NPs in the polymer network as comparison. In the simulations, two rigid ellipsoidal NPs with different aspect ratios (2:1 and 3:1) were used, which had the same volume as the spherical NPs. The results indicated that the diffusivities of the rigid NPs (one spherical NP and two ellipsoidal NPs) were influenced by the aspect ratio of the NPs. The ellipsoidal NPs diffused faster than the rigid spherical NPs. We further found that the diffusivity of spherical semi-elastic NPs lied between those of the rigid spherical NPs and rigid ellipsoidal NPs with aspect ratio 2:1 (see Fig. R1). These results further provided evidence that the semi-elastic NPs would change its shape from a sphere to an ellipsoid during diffusion. We have added these results in our revised manuscript and Supplementary Information.

Figure. R1 | Ddiffusivity of different types of NPs in the polymer network. (a) The MSDs for different types of NPs and (b) the diffusivities for different types of NPs. The red, blue, dark

cyan, and purple lines represent semi-elastic NPs, hard spherical NPs, rigid ellipsoidal NPs with aspect ratio 2, and rigid ellipsoidal NPs with aspect ratio 3, respectively.

Line 467 mentions "Time-averaged mean squared displacement". It should be "Particle-averaged mean squared displacement". The equations for MSD at time t and diffusivity (lines 469 and 553) should indicate somehow that an average is taken over all of the particles.

Response: In response to this comment, we have corrected the sentence in the revised manuscript as: "Particle-averaged mean squared displacement (MSD) and effective diffusivities (D_{eff}) were calculated using the following equations:

$$\text{MSD}(t) = \frac{\sum [(x_t - x_0)^2 + (y_t - y_0)^2]}{n}$$
$$D_{\text{eff}} = \frac{\sum [\text{MSD}_t / (4t)]}{n}$$

For fitting the simulation MSDs, it is stated that the MSDs were fit starting from time 0 (line 563). Generally, there is a short time over which the MSD is not expected to be linear. Therefore, fitting should usually not start from time 0. This nonlinearity at short times can be seen in Figure 6 d. Given the length of the trajectories, throwing away this short time will not have a large effect on the results. However, one way to determine this time is to plot the derivative of the log of MSD with respect to the log of time. Once that begins fluctuating around a value of 1, then the MSD has become linear. See the attached plot, where the nonlinear part lasts between 10 and 15 ps and the fitting should not be started until after that.

Response: The reviewer is right that the fitting should not start from time 0. In our simulations, the total simulation time was $2 \times 10^6 \tau$. After about $8 \times 10^5 \tau$, the MSD calculations were performed. The fitting of the diffusivity D was calculated by linearly fitting the MSD versus time from $1.3 \times 10^6 \tau$ to $1.8 \times 10^6 \tau$. The slope of the fitting line is denoted by k ; thus, $D = k/6$.

We thank the reviewer for her/his positive evaluation of our manuscript and the constructive comments/suggestions which have helped us improve the manuscript.

Reviewer #2 (Remarks to the Author):

In this paper, the authors tested their hypothesis that particle diffusion in biological hydrogels would be dependent upon particle rigidity. In particular, they formulated core-shell PLGA-lipid NPs possessing a range of rigidity, as determined by Young's moduli, and demonstrated that semi-elastic NPs were most efficient at mucus and tumor penetration. This paper is relevant to the readership of the journal and easy to read, and experiments are generally well designed and executed. However, the connection between their experimental observation and data interpretation is relatively loose and there are a few critical issues to be addressed, as listed below, prior to its consideration for publication.

Response: We thank the reviewer for her/his positive evaluation of our manuscript and constructive comments/suggestions which have helped us improve the manuscript substantially.

Specific comments:

1. The authors interpreted their experimental observation largely based on MD simulation, but the connection does not seem tight. Specifically, there are many assumptions in the simulation without clear justifications and/or relevant references. For example, based on what rationale the authors set the relative dimensions of particles and mesh pore sizes in the simulation to be 10:16? How the degree of polymer affinity was selected to model biological hydrogels (i.e. mucus vs. tumor, etc.)? Overall, the authors must provide clear rationales for the use of different numbers, equations and theories with appropriate references.

Response: It is generally challenging to construct an atomistic model for simulating diffusion processes, especially in view of the complexity of mucus components and structures. To address the reviewer's comment, we constructed a full-atomistic

simulation model to extract the affinity of the polymer. Based on the extracted parameter, we adopted a coarse-grained model to investigate the diffusion process, which we have successfully used before to study the diffusion of rigid NPs in the mucus [Nano Lett. 2016, 16 (11): 7176]. Such model has also been used by other groups to investigate the diffusion of NPs in polymer nanocomposites [Phys. Rev. Lett. 2014, 112: 108301].

It has been reported that the mesh size of mucus and ECM varies in a wide range of 10-1000 nm^{1, 2, 3, 4, 5, 6}. In our experiment we found that the average pore size of the mucus was about 100-240 nm⁷, and the hydrodynamic diameter of the NPs was about 200 nm, which is close to the size of the mucus pore. To mimic experiment, we modeled the dimensions of particles and mesh pore sizes in the simulation to be 10:16. In response to the reviewer's concern, we have considered decreasing the pore size to 14 σ , in which case all the NPs were trapped in the network oscillating within one grid cell (Fig. R2). We also increased the pore size to 18 σ , and observed an increase in the diffusivity due to loosened restriction of the polymer network. Still, the semi-elastic NPs diffused faster than the soft and hard NPs (Fig. R2).

Figure R2 | Effect of the pore size of the polymer network on the diffusion of NPs in a model system. The (a) MSDs and (b) diffusivities for three types of NPs when the pore size of the polymer network is tuned from 14 σ to 18 σ gradually. The green, red, and blue lines represent soft, semi-elastic, and hard NPs, respectively.

To properly reflect the polymer affinity, we conducted additional atomistic simulations in which the interactions between the lipid-coated NPs / F127-coated NPs

and polymer chains were extracted as follows.

In our experiments, the liposomes were coated with F127 molecules. To obtain the interaction energy between NPs and mucin fibers, we constructed two model systems: (1) one bilayer interacting with one mucin glycoprotein chain; and (2) one long $-(\text{OCH}_2\text{CH}_2)_n\text{-OH}$ coil interacting with one mucin glycoprotein chain. From the first model we could extract the affinity between the bare liposome and mucin fibers, while from the second model we could extract the affinity between the F127-coated liposome and mucin fibers. Our results showed that the affinity strength between lipid bilayer and mucin glycoprotein is about $-72.51\text{kJ} \cdot \text{mol}^{-1} \cdot \text{nm}^{-2}$, much larger than that between $-(\text{OCH}_2\text{CH}_2)_n\text{-OH}$ chains and mucin glycoprotein (about $-2.29\text{kJ} \cdot \text{mol}^{-1} \cdot \text{nm}^{-2}$).

We then set the parameter $\varepsilon_{ij} = 0.1$ in our coarse grained models, in which case the interaction strength between the NPs and polymer chain is about $-5.17\text{kJ} \cdot \text{mol}^{-1} \cdot \text{nm}^{-2}$, compared to the value of $-2.29\text{kJ} \cdot \text{mol}^{-1} \cdot \text{nm}^{-2}$ from atomistic simulations. The atomistic simulation model, method and results were summarized in Supplementary Information. The interaction parameters between the polymer network and NPs were also tuned in a wide range from $\varepsilon_{ij} = 0.02$ to $\varepsilon_{ij} = 0.2$. Here the lower bound of interaction represents the extreme case in which the network has almost negligible affinity with the NPs. This case is not representative of real mucus, where there exist multiple interactions such as hydrogen bonding, electrostatic interaction, Van Der Waals interaction, ionic interaction etc. The results showed that in this case the soft NPs diffused faster than the intermediate and hard NPs (Fig. R3(a-b)). The upper bound of interaction represents another extreme case in which the network has strong interaction with the NPs. It was seen that all the NPs were trapped within the grid cell and cannot freely move (Fig. R3(g-h)). This case is at odds with our experimental observation that the NPs could move from one grid cell to another. When the parameter was tuned to an intermediate one, $\varepsilon_{ij} = 0.05$ or $\varepsilon_{ij} = 0.1$ for instance, the

NPs could move and the semi-elastic NPs diffused the fastest (Fig. R3(d-e) and Fig. 6(d-e)). Based on these comparisons, we selected the intermediate value to describe the interaction. The results were summarized in the Supplementary Information.

Figure R3 | Effect of interaction parameters on the diffusion of NPs in a model system. (a)-(c) The MSDs, diffusivities, and contact number for three types of NPs when the interaction parameter between polymer network and NPs is relatively small $\varepsilon_{ij} = 0.02$. (d)-(f) The MSDs, diffusivities, and contact number for three types of NPs when the interaction parameter between polymer network and NPs is of an intermediate value $\varepsilon_{ij} = 0.05$. (g)-(i) The MSDs, diffusivities, and contact number for three types of NPs when the interaction parameter between polymer network and NPs is of a relative large value $\varepsilon_{ij} = 0.2$. The green, red, and blue lines represent soft, semi-elastic, and hard NPs, respectively.

2. While the diffusivity values for soft and hard NPs are comparable in rat intestinal mucus (Figure 2b), MD simulation suggests that hard NPs move much faster than soft NPs (although statistical analysis is not provided) (Figure 6). This discrepancy again

raise a concern for using the simulation to interpret the authors' experimental data.

Response: Thank the reviewer for pointing out this issue. The discrepancy between experiment and simulation is due to the selection of the interaction parameter between the NPs and the polymer, which could influence the diffusivity of NPs greatly. As seen in Fig. R3, the diffusivity of soft NPs is even larger than those of hard and semi-elastic NPs when the interaction is small. Only when the interaction is in an appropriate range, could the semi-elastic NPs attain the highest diffusivity. In response to the reviewer's concern, in our revised manuscript, we have determined the interaction parameters as $\varepsilon_{ij} = 0.1$ (as opposed to qualitatively adopting an intermediate value as we did previously) from the full-atom simulations. Once this is done, the MD results are now much closer to the experimental results (Fig. R4).

Figure R4 | Snapshots and trajectories of NPs in mucus fibers. (a) The original construction of NP. (b) A snapshot of the polymer network (blue and yellow) and NPs during the simulation. (c) Typical centroid trajectories of NPs in simulations. (d) The representative MSD values of the three types of NP in a polymer network with a mesh size of 16σ . (e) The diffusivities of NPs in the polymer network. (f) The number of beads in each NP that came into contact with the polymer network. The blue, red, and green labels represent hard, semi-elastic, and soft NPs, respectively.

3. The authors claimed that soft NPs exhibited lowest diffusivity as they “deformed excessively and attached themselves to the polymer”, which indicates that particle diffusion is largely hindered by adhesive interactions rather than flexibility itself.

However, it has been previously reported that NPs can effectively resist adhesion to mucus when their surfaces are densely coated with F127. This raises a question whether their NPs with different rigidities possess equally well shielded surfaces; while z-potential measurement is one way to characterize particle surfaces, exhibiting similar z-potentials among different NP population does not ensure similar/identical NP surface properties. It is conceivable that NPs may efficiently move in a biological hydrogel regardless of rigidity, if their surfaces are well shielded unless NPs are extremely elongated to intertwine with gel fibers. In addition, what would be the role of deformability of NPs on their viscous drag?

Response: We thank the reviewer for bringing up this issue. Due to their low rigidity, the soft NPs are apt to undergo deformation, and the contact area between NPs and hydrogel could increase after deformation, resulting in increased adhesion. As a result, the soft NPs exhibit the lowest diffusivity. So the increased adhesive interactions are attributed to the flexibility. We have rewritten the statements in the revised manuscript: *“We observed that during transportation, due to their low rigidity, the soft NPs are apt to undergo deformation which could increase the contact area between NPs and hydrogel resulting in increased adhesion and the lowest diffusivity.”*

In addition, we agree with the reviewer that exhibiting similar z-potentials among different NP population does not ensure similar/identical NP surface properties. So we have elected to use liposomes as the shells of the three types of NPs (soft, semi-elastic and hard NPs) in our study, as shown in Fig. 1b. The compositions of the liposomal shells were the same, and all the shells represented phospholipid bilayer with a width of 8 nm (Fig. 1b). Combining z-potentials, liposomal compositions and Cryo-TEM results, we think the fabricated NPs should have similar NP surface properties.

It has been previously reported that sufficient PEG surface density must be attained for NPs to rapidly diffuse through mucus, but with only minor additional increase in MSD for NPs with more PEG content⁸. Similarly, the density of F127 displayed on the surface of NPs could dramatically influence the interactions between NPs and mucin fibers. We have previously synthesized the PLGA-lipid NPs with varying F127 contents (5, 10, and 20 wt %, referred to as PLGA₇₀-Lip-F127_{5%},

PLGA₇₀-Lip-F127_{10%}, and PLGA₇₀-Lip-F127_{20%}). All NPs were approximately 200 nm and monodispersed (Table R1). MPT were then used to observe the transport dynamics of NPs in freshly obtained rat intestinal mucus. As shown in Fig. R5, there is only a minor additional increase in MSD of NPs as F127 content increased from 5% to 10%, indicating that 5% F127 density was sufficient for uniform penetration in biological hydrogel. The surfaces of NPs we synthesized were well shielded, and NPs are not elongated to intertwine with gel fibers.

Table R1 | Comparison of mean diffusivity of nanoparticles with different content of F127.

Nanoparticle	Hydrodynamic diameter (nm)	PDI	Zeta potential (mV)	Diffusivity ($\mu\text{m}^2/\text{s}$)	Ratio of diffusivity (NPs/liposome)
Lip	204.9 \pm 6.9	0.133	-11.3 \pm 1.9	0.019	/
Lip-F127 _{5%}	209.4 \pm 3.8	0.123	-5.8 \pm 1.9	0.168	8.8
PLGA ₇₀ -Lip-F127 _{5%}	203.6 \pm 3.9	0.149	-4.8 \pm 0.6	1.852	97.5
PLGA ₇₀ -Lip-F127 _{10%}	220.2 \pm 5.7	0.189	-5.7 \pm 2.4	1.895	99.7
PLGA ₇₀ -Lip-F127 _{20%}	226.0 \pm 4.9	0.201	-3.7 \pm 2.8	0.919	48.4

Figure R5 | Transport of NPs with various F127 contents (0, 5, 10, and 20% F127) in rat intestinal mucus. MSD and distributions of the logarithms of effective diffusivities (D_{eff}) of nanoparticles with different content of F127.

The reviewer also raised a question about the viscous drag. Traditionally, diffusion of particle in a continuous medium is described by the Stoke-Einstein

relation, which shows that the diffusivity is inversely proportional to the solvent viscosity and the size of the particle. In a viscous flow, the NP could change its shape if it is deformable, thus the size of NP could also change, leading to change in diffusivity. Thus in a continuous medium, deformability could have an important effect on viscous drag. In a polymer solution, as pointed out by de Gennes, the continuum fluid assumption is in question because flow around NPs is no longer captured by the solvent viscosity, rather the local friction (due to the polymer) experienced by the NP becomes the key [Eur. Phys. J. E: 2000, 1: 93]. Hence we have focused on the interaction between NPs and polymer in our study.

4. It is unclear how the rotational behaviors of semi-elastic NPs facilitate their movements in a biological hydrogel. The authors must provide more detailed and precise interpretation of their observations beyond speculation. In addition, the range of Young's moduli tested in this study is relative narrow (i.e. 5 – 110 MPa) and the authors are claiming that doubling the value (i.e. semi-elastic to hard) significantly impacts the NP mobility in hydrogels. However, multiple independent groups have shown that NPs possessing over an order of magnitude greater Young's moduli, such as polystyrene beads, efficiently penetrate mucus and tumor tissues when those particles are small enough and capable of resisting adhesion to the mucus.

Response: We thank the reviewer for this insightful comment. Previously, we have presented evidence that, compared with spheres, cylindrical/ellipsoidal NPs display superior diffusion and penetration patterns in biological hydrogels, such as the intestinal mucus⁹. Molecular simulations and stimulated emission of depletion (STED) microscopy revealed that the rotational diffusion of ellipsoid NPs within the complex mesh structure plays an active role in such diffusion enhancement. Once semi-elastic NPs deform into ellipsoids because of their moderate stiffness, this thin shape could further facilitate the transport of the semi-elastic NPs. We have added more detailed and precise interpretation in the revised manuscript. *“MD simulations and super-resolution microscopy” part, “We have recently applied molecular simulations and stimulated emission of depletion (STED) microscopy to reveal that the movement*

of ellipsoid NPs in a mesh structure consists of two parts, rotation around the polymers and translational diffusion, which contributes to their superior diffusivity over spherical NPs. The elongated shape enabled the ellipsoidal NPs to avoid becoming trapped by the polymer and therefore achieve higher diffusivity than their spherical counterparts.”

In the current study, we have shown that PLGA-PEG NPs, which possess over an order of magnitude greater Young's moduli (approximately 1 GPa)¹⁰, exhibit good mucus diffusion capacity (Supplementary Fig. S9 and Table S3). On a time scale of 1 s, the <MSD> value of PLGA-PEG NPs was observed to be approximately 1.2671, which is consistent with previous result⁸. However, compared with PLGA-PEG, the semi-elastic NPs also displayed superior mucus diffusion. So the range of Young's moduli tested here is not relative narrow.

We agree with the reviewer that NPs possessing over an order of magnitude greater Young's moduli could efficiently penetrate mucus and tumor tissues when those particles are small enough and capable of resisting adhesion to the mucus. However, extremely small NPs are inefficiently endocytosed^{11, 12}. This can hinder successful delivery of targeted NPs in tissues, which needs to overcome both the biological hydrogel barrier and cellular barrier. Therefore, here we explored the effect of rigidity in simultaneously overcoming the biological hydrogel barrier and cellular barrier.

5. The author claimed that their NPs with moderate rigidity showed unprecedented diffusivity through mucus compared to any known synthetic NPs, which is an overstatement. The authors formulated some possible MPP formulations and showed that their semi-elastic NPs exhibited greater diffusivity values, but it is unclear whether those control particles were carefully made to efficiently penetrate mucus. For example, z-potentials of PLGA-PEG NPs are markedly more negative (-15 mV) compared to those of core-shell NPs (-5 mV), suggesting that PLGA-PEG are possibly not well shielded by PEG. In addition, it is unclear whether the differences among different formulations are statistically significant.

Response: We thank the reviewer for bringing up this issue. In response to the reviewer's comment, we have rewritten the concerned statement in our revised manuscript: "*we demonstrate that (poly(lactic-co-glycolic acid) (PLGA) core)-(lipid shell) NPs with moderate rigidity display enhanced diffusivity through mucus compared with some synthetic mucus penetration particles (MPPs)*".

We have carefully fabricated the control particles according to previous work^{7, 8} in the revised manuscript, and now the z-potential of PLGA-PEG NPs is approximate -5 mV. We have conducted the "Mucus penetrating particle (MPP) transport in rat intestinal mucus ex vivo" again, and the results are comparable to previously reported data. On a time scale of 1 s, the <MSD> value of PLGA-PEG NPs was approximately 1.2671, which is consistent with the reported result⁸.

In addition, we have added a statistical analysis for the different formulations.

6. Several necessary details are missing throughout the method section, which makes it hard to precisely gauge the validity of this study and to reproduce the observations here by an independent group. For example, why two different pluronics (i.e. F68 and F127) were used for the formulation of core-shell NPs? how were the Young's moduli of NPs calculated? What was the properties of intestinal mucus collected from rodents (e.g. mucin concentration, etc.) and how many animals were sacrificed to collect mucus? What was the concentration of NPs added to mucus for particle tracking studies? How many movies were taken and analyzed? How the MPP-control particles were made; did the authors strictly follow published methods of formulation and characterization and/or collaborate with relevant groups? How the particle tracking data was processed/averaged? Was the presence of mucus confirmed in the cell culture studies? Were qualitative image-based studies repeated to corroborate the authors' findings or could the authors conduct image-based quantification?

Response: We thank the reviewer for these constructive criticisms. We have added detailed methods in our revised manuscript.

It is reported that pluronic F127 is a good candidate for decorating NPs to enhance mucus diffusivity^{13, 14, 15, 16, 17, 18}. We adopted pluronic F127 to modify the

shell of the NPs (liposome) to facilitate their mucus penetration capacity. For the fabrication of PLGA cores, we found that pluronics F68 was more suitable to make the particle size controllable. So we applied these two different pluronics for the formulation of core-shell NPs.

All images of NPs obtained from AFM were processed by the software NanoScope Analysis (Bruker), and the Young's modulus was calculated by processing the images of NPs in PeakForce DMT mode (n=3). The detailed computation principle and method are shown in Fig. R6.

Figure R6 | Determination of modulus from force curve — DMT model.

For mucus collection, we have adopted methods reported by the Hanes group^{19, 20, 21, 22}. Briefly, the small intestine was excised after euthanizing the rats, and approximately 1.5-2 mL of mucus from each fasted rat was collected. The properties of intestinal mucus collected from rodents have been reported in our previous work⁹, which is in consistent with previous reports^{5, 23, 24}. The average size of the mucus pore was approximately 200 nm, and the majority of pores were 100-240 nm in diameter. In this study, ten rats were sacrificed to collect mucus for all particle tracking studies. The concentration of NPs added to mucus for particle tracking studies was 100 µg/ml in PBS, and fifteen movies were taken and analyzed for each NP. We have added these supplements in the revised manuscript.

We have adopted the reported methods by the Hanes group^{7, 8} to fabricate the MPP-control particles in the revised manuscript, and we have properly cited the literatures in the manuscript.

The particle tracking movies were processed by the software ImageJ, which could transform the variation of coordinate position into motion trajectories of particles. We then collected the tracking data of 300 particles followed by calculating time-averaged mean square displacement (MSD) and effective diffusivities (D_{eff}).

Prior to the experiment, E12 cells were incubated for one week to ensure the formation of mucus. We have applied a method reported previously²⁵ to ensure the distribution of mucus on the apical surface of HT29-MTX-E12 cells, fluorescence staining (Alexa Fluor 488 labeled wheat germ agglutinin) were applied. The mucus layer were stained with Alexa Fluor 488 labeled wheat germ agglutinin (10 $\mu\text{g/ml}$) for 10 min at 37 $^{\circ}\text{C}$. The cells were then stained with DAPI for nuclei identification and visualized using CLSM. The result is consistent with the reported data²⁵. And it is visible that homogeneous mucus has covered on the surface of cell monolayer (Fig. R7).

Figure. R7. Confocal microscope images of E12 cells layers. Green: mucus covering the cell surface stained with Alexa Fluor 488 labeled wheat germ agglutinin. Blue: cell nucleus stained with DAPI. Scale bar: 10 μm .

7. The authors used student's t-test for statistical analyses, but multiple comparisons should be made with ANOVA. In addition, statistical analysis is missing throughout the manuscript, particularly for particle tracking studies.

Response: In response to the reviewer's suggestion, we have updated the methods for statistical analyses. We have also added the statistical analysis for particle tracking studies.

8. In Figure 4, it is unclear whether semi-elastic NPs indeed provided improved distribution in mucus and tumor tissue. While the overall fluorescence intensity was greater in the intestinal lumen for semi-elastic NPs, all NPs seemed to distribute near the epithelial surfaces, indicating efficient mucus penetration. Likewise, while more semi-elastic NPs were accumulated in tumor compared to other NPs following systemic administration, it may be due to the difference in their blood pharmacokinetics rather than their abilities to penetrate tumor tissue. Comparing the NP distribution following local administration would be a more relevant experiment to demonstrate superior ability of semi-elastic NPs to diffuse in tumor tissues.

Response: Thank the reviewer for bringing up this issue. We understand the reviewer's concern about the superiority of semi-elastic NPs in mucus and tumor tissue penetration. The NPs synthesized here (soft, semi-elastic, and hard NPs) were all modified with F127. And it is reported that F127 could facilitate the mucus penetration of NPs^{13, 14, 15, 16, 17}. So it is true that all NPs could penetrate the mucus to a certain extent, which is also consistent with previous results^{13, 14, 15, 16, 17}. However, for the semi-elastic NPs, strong fluorescence intensity was observed in the intestinal lumen and near the epithelial surfaces, indicating the superior diffusivity capacity of the semi-elastic NPs.

For the study of NP penetration in tumor tissue, we thank the reviewer for this insightful suggestion. We agree that the rigidity of NPs would affect their blood pharmacokinetics. So we have adopted the suggestion and conducted an experiment in which all the NPs were locally administrated. As shown in Fig. R8a, the soft and hard NPs displayed weak fluorescence from the tumor at 6 h post-injection, which further reduced with time. In sharp contrast, the semi-elastic NPs exhibited a strong fluorescence signal, indicating that semi-elastic NPs could be efficiently internalized into the tumor and stably accumulated within cancer cells. Tumors were then excised from the mice at 6 h after peritumoral injection and sliced for imaging. The semi-elastic NPs were detected throughout the tumors, whereas other NPs were detected only in the perivascular sections of tumors (Fig. R8b).

Figure. R8 | (a) *In vivo* imaging of BxPC-3 and HPSC tumor-bearing mice taken at the indicated time points before and after peritumoral injection of IR783-labeled soft, semi-elastic, or hard NPs. (b) Distribution of DiI-labeled softer, semi-elastic, and harder NPs in tumor slices of BxPC-3 & HPSC tumor xenografts 6 h after peritumoral injection. The cell nuclei were counterstained with DAPI. The tumor vessels were labeled with anti-CD31 antibody. Scale bars: 50 μm. Softer: Lip-F127_{5%} NPs; Semi: PLGA₇₀-Lip-F127_{5%} NPs; Harder: PLGA₁₆₀-Lip-F127_{5%} NPs.

9. Why soft NPs retained much shorter in tumor *in vivo* compared to other NPs if they readily adhere to the tumor tissue as suggested by the authors (Figure 6b)?

Response: The factors that influence the tumor retention include penetration and cellular uptake. Since the soft NPs showed both the weakest biological hydrogel penetration capacity and cellular uptake ability, they just adhered to the surface of tumor tissue, and most of the soft NPs could not penetrate deep into the tumor tissue. Thus, one could suppose that the majority of soft NPs were cleared through the blood flow. As a result, they exhibited much shorter retention in tumor *in vivo* compared with other NPs.

10. Rhodamine B does not “label” the water, but visualize the volume occupied by water.

Response: We thank the reviewer for pointing out this typo, and we have corrected the sentence as: “*To characterize the amounts of interfacial water inside the hybrid NPs, we used ethyl rhodamine B to visualize the volume occupied by the water encapsulated by the NPs according to a previously reported method*”.

Reviewer #3 (Remarks to the Author):

In their paper “Rapid Transport of Deformation-Tuned Nanoparticles across Biological Hydrogels and Cellular Barriers”, the authors show that particles of different rigidities but identical surface chemistry and size can have very different diffusion properties within biological hydrogels. The experiments are straightforward and convincing, notwithstanding some quibbles with Fig. S4 described below. Overall, it is my recommendation that the manuscript be accepted for publication with minor revisions.

Response: Thank you for your positive assessment of our work.

More specifically, to me it seems like the results of this paper suggest that a nanoparticle with a 70nm solid core and 200nm shell diffuses essentially as though it were 70nm solid nanoparticle, because it can deform to slip through a 70nm pore by presenting a 70nm cross section. Or more generally, a PLGA_x-Lip-F127_{5%} particle could diffuse essentially like a solid nanoparticle of diameter x, except that for small PLGA cores this trend stops holding because the liposome can deform around gel polymers. If this is true, then the natural comparison looking at nanoparticle size changes in Fig. S4 would be to compare PLGA₇₀-Lip-F127_{5%} to hard, rather than soft, smaller nanoparticles. As it is, I am not entirely sure what the motivation for the Fig. S4 experiment is.

Response: Thank the reviewer for these insightful comments. The motivation for the Fig. S4 experiment is to explore whether tuning the rigidity of liposomes is more effective for biological hydrogel penetration compared to changing the size. We have rewritten the sentence in the revised manuscript as: “*which indicated that elasticity might be a more important factor than size for the transport of liposome across mucus.*”

In addition, we have previously showed that ellipsoid NPs with hydrodynamic diameter of 200 nm displayed superior diffusion and penetration patterns in biological

hydrogels compared to the rigid sphere NPs with hydrodynamic diameter of 80 nm⁹. Molecular simulations and stimulated emission of depletion (STED) microscopy revealed that the rotational diffusion of ellipsoid NPs within the complex mesh structure plays an active role in such diffusion enhancement. In this study, we presented evidences that semi-elastic NPs could change their shape to ellipsoidal ones, and this thin shape further facilitated their diffusion in biological hydrogels.

At any rate, the discussion may benefit from clarifying the advantages of semi-elastic nanoparticles in potential applications. Compared to hard nanoparticles, this paper shows that semi-elastic particles have a lower effective diameter for diffusion (which is good!), but also have a lower uptake (which is bad!) possibly also essentially due to a smaller effective diameter, so semi-elastic particles are basically equivalent to smaller hard particles. Is the idea just that you can fit more stuff in the semi-elastic particles, which can be bigger than the equivalent smaller, solid particle? That's interesting! And could perhaps be stated explicitly. But overall, the benefits of semi-elastic larger nanoparticles over smaller, hard nanoparticles seem somewhat marginal. However, it appears that adding a solid core to (soft) liposomes would be helpful on many levels, from diffusion to uptake. That's very interesting. So it seems as though the utility of this paper will come largely from applications where a liposome, rather than a solid nanoparticle, is desired for the biological effect. If this is true it is potentially worth stating directly.

Response: We again thank the reviewer for these constructive comments. It is true that the semi-elastic particles could load more stuff, which would benefit drug delivery. We have adopted the reviewer's suggestion and added some discussion in the revised manuscript: *"In addition, adding a solid core with suitable size to (soft) liposomes would be helpful on many levels, from diffusion to uptake, and further for drug delivery especially for overcoming the multi-biological barriers."*

Some minor comments:

- I think the capitalization on the title is off, the copy editor can presumably confirm

this

- Line 672: it's the Hanes group, not the Justin group

Response: We thank the reviewer for pointing out this typo, and we have corrected the sentence: "*the Hanes Group has fabricated mucus penetrating particles (MPPs)*".

- Line 175: I think the word "frequency" should be deleted here: "Distributions of the logarithms of individual..." seems fine.

Response: We thank the reviewer for pointing out this typo, and the word "frequency" has been deleted in the revised manuscript.

- Line 366: to stay consistent it should be "Fig. S5" (the period was left out)

Response: We have added the period in the revised manuscript.

- Figures S4 and S5 are mentioned after Fig. S6-S8, so they should be reordered.

Response: Thank the reviewer for bringing up this issue. We have reordered the Figures S4 and S5. The Figures S4 and S5 are now Figures S8 and S9, respectively.

- Figure S9 is never referenced in the main text

Response: Thank you for point out this issue. Actually, this figure was used to verify the robust of our coarse-grained model. We have added the following sentence in the modified manuscript to explain it. "*To verify the robust of our simulation model, we also tuned the interaction parameter between the polymer network and NPs from $\varepsilon_{ij} = 0.02$ to $\varepsilon_{ij} = 0.2$ and the pore size of the network from 14σ to 18σ in the simulations, the results were summarized in Supplementary Information and Fig. S12 and S13.*"

Reference:

1. Netti PA, Berk DA, Swartz MA, Grodzinsky AJ, Jain RK. Role of extracellular matrix assembly in interstitial transport in solid tumors. *Cancer Res* 2000, **60**(9): 2497-2503.
2. Witten J, Ribbeck K. The particle in the spider's web: transport through biological hydrogels. *Nanoscale* 2017, **9**(24): 8080-8095.
3. Chauhan VP, Stylianopoulos T, Boucher Y, Jain RK. Delivery of molecular and nanoscale medicine to tumors: transport barriers and strategies. *Annu Rev Chem Biomol Eng* 2011, **2**: 281-298.
4. Zhang SS. [Treatment of chronic pleural empyema by filling with the pedicled omentum: report of 10 cases]. *Zhonghua Jie He He Hu Xi Ji Bing Za Zhi* 1985, **8**(4): 218-220, 255.
5. Lai SK, Wang YY, Hida K, Cone R, Hanes J. Nanoparticles reveal that human cervicovaginal mucus is riddled with pores larger than viruses. *Proc Natl Acad Sci U S A* 2010, **107**(2): 598-603.
6. Birjiniuk A, Billings N, Nance E, Hanes J, Ribbeck K, Doyle PS. Single particle tracking reveals spatial and dynamic organization of the E. coli biofilm matrix. *New J Phys* 2014, **16**(8): 085014.
7. Yu T, Chan KW, Anonuevo A, Song X, Schuster BS, Chattopadhyay S, *et al.* Liposome-based mucus-penetrating particles (MPP) for mucosal theranostics: demonstration of diamagnetic chemical exchange saturation transfer (diaCEST) magnetic resonance imaging (MRI). *Nanomedicine* 2015, **11**(2):401-405.
8. Xu Q, Ensign LM, Boylan NJ, Schon A, Gong X, Yang JC, *et al.* Impact of Surface Polyethylene Glycol (PEG) Density on Biodegradable Nanoparticle Transport in Mucus *ex Vivo* and Distribution *in Vivo*. *ACS Nano* 2015, **9**(9):9217-9227.
9. Yu M, Wang J, Yang Y, Zhu C, Su Q, Guo S, *et al.* Rotation-Facilitated Rapid Transport of Nanorods in Mucosal Tissues. *Nano Lett* 2016, **16**(11): 7176-7182.
10. Park JH, Allen MG, Prausnitz MR. Biodegradable polymer microneedles: fabrication, mechanics and transdermal drug delivery. *J Control Release* 2005, **104**(1):51-66.
11. Gao H, Shi W, Freund LB. Mechanics of receptor-mediated endocytosis. *Proc Natl Acad Sci U S A* 2005, **102**(27):9469-9474.
12. Jiang W, Kim BY, Rutka JT, Chan WC. Nanoparticle-mediated cellular response is size-dependent. *Nat Nanotechnol* 2008, **3**(3):145-150.
13. Chen D, Xia D, Li X, Zhu Q, Yu H, Zhu C, *et al.* Comparative study of Pluronic((R)) F127-modified liposomes and chitosan-modified liposomes for mucus penetration and oral

absorption of cyclosporine A in rats. *Int J Pharm* 2013, **449**(1-2):1-9.

14. Ensign LM, Lai SK, Wang YY, Yang M, Mert O, Hanes J, *et al.* Pretreatment of human cervicovaginal mucus with pluronic F127 enhances nanoparticle penetration without compromising mucus barrier properties to herpes simplex virus. *Biomacromolecules* 2014, **15**(12):4403-4409.
15. Yang M, Lai SK, Yu T, Wang YY, Happe C, Zhong W, *et al.* Nanoparticle penetration of human cervicovaginal mucus: the effect of polyvinyl alcohol. *J Control Release* 2014, **192**: 202-208.
16. Yang M, Yu T, Wang YY, Lai SK, Zeng Q, Miao B, *et al.* Vaginal delivery of paclitaxel via nanoparticles with non-mucoadhesive surfaces suppresses cervical tumor growth. *Adv Healthc Mater* 2014, **3**(7): 1044-1052.
17. Li X, Guo S, Zhu C, Zhu Q, Gan Y, Rantanen J, *et al.* Intestinal mucosa permeability following oral insulin delivery using core shell corona nanolipoparticles. *Biomaterials* 2013, **34**(37): 9678-9687.
18. Li X, Chen D, Le C, Zhu C, Gan Y, Hovgaard L, *et al.* Novel mucus-penetrating liposomes as a potential oral drug delivery system: preparation, in vitro characterization, and enhanced cellular uptake. *Int J Nanomedicine* 2011, **6**: 3151-3162.
19. Ensign LM, Henning A, Schneider CS, Maisel K, Wang YY, Porosoff MD, *et al.* Ex vivo characterization of particle transport in mucus secretions coating freshly excised mucosal tissues. *Mol Pharm* 2013, **10**(6): 2176-2182.
20. Maisel K, Ensign L, Reddy M, Cone R, Hanes J. Effect of surface chemistry on nanoparticle interaction with gastrointestinal mucus and distribution in the gastrointestinal tract following oral and rectal administration in the mouse. *J Control Release* 2015, **197**: 48-57.
21. Date AA, Hanes J, Ensign LM. Nanoparticles for oral delivery: Design, evaluation and state-of-the-art. *J Control Release* 2016, **240**: 504-526.
22. Ensign LM, Cone R, Hanes J. Oral drug delivery with polymeric nanoparticles: the gastrointestinal mucus barriers. *Adv Drug Deliv Rev* 2012, **64**(6): 557-570.
23. Bajka BH, Rigby NM, Cross KL, Macierzanka A, Mackie AR. The influence of small intestinal mucus structure on particle transport ex vivo. *Colloids Surf B Biointerfaces* 2015, **135**: 73-80.
24. Kirch J, Schneider A, Abou B, Hopf A, Schaefer UF, Schneider M, *et al.* Optical tweezers reveal relationship between microstructure and nanoparticle penetration of pulmonary mucus. *Proc Natl Acad Sci U S A* 2012, **109**(45): 18355-18360.
25. Liu M, Zhang J, Zhu X, Shan W, Li L, Zhong J, *et al.* Efficient mucus permeation and tight

junction opening by dissociable "mucus-inert" agent coated trimethyl chitosan nanoparticles for oral insulin delivery. *J Control Release* 2016, **222**: 67-77.

Reviewers' Comments:

Reviewer #1:

Remarks to the Author:

After looking at the responses of the authors and the changes in the manuscript and supplement, I believe the authors addressed the issues brought up by the reviewers and recommend this work for publication.

Reviewer #2:

Remarks to the Author:

The authors made significant effort to carefully address the raised concerns. Additional clarification/elaboration throughout the manuscript as well as complementary experiments improved the quality and reliability of this manuscript. I would like to make a couple of subtle remarks that do not necessarily affect the final conclusion of this work.

1. I agree that particle deformation can increase the contact area between NPs and hydrogel, but the increase in the contact area yields elevated adhesion only if the particle surface is inherently adhesive to a certain degree. If the particle surface is completely resistant to any adhesion, deformation may not result in increased adhesion, and thus reduced diffusion rates, unless particles are physically entangled with hydrogel. As an additional possibility, particle deformation that occurs during their transport in hydrogel may lead to alteration in the surface property that renders particle more adhesive than its original state.

2. The authors made a statement that extremely small NPs are inefficiently endocytosed, which I believe is an over-generalization and the dependence of particle size on endocytosis varies with type of cells and range of particle sizes (i.e. how "extremely" small?). Prior reports demonstrate that ~100 nm NPs produced to avoid adhesion to mucus or other biological hydrogels are small enough to pass through hydrogel pores and these particles are certainly in a good range for efficient endocytosis.

Reviewer #3:

Remarks to the Author:

The authors addressed the main criticism raised by the reviewers and thereby substantially strengthened the manuscript. I can now recommend it for publication.

Response to Review

We are grateful to the Editorial Office and reviewers for their constructive comments and suggestions. We have carefully revised the paper according to the reviews' comments. The changes are highlighted in the revised manuscript. The following are our detailed responses:

REVIEWERS' COMMENTS:

Reviewer #1 (Remarks to the Author):

After looking at the responses of the authors and the changes in the manuscript and supplement, I believe the authors addressed the issues brought up by the reviewers and recommend this work for publication.

Response: We thank the reviewer for his/her positive evaluation of our manuscript and constructive comments/suggestions which have helped us improve the manuscript substantially.

Reviewer #2 (Remarks to the Author):

The authors made significant effort to carefully address the raised concerns. Additional clarification/elaboration throughout the manuscript as well as complementary experiments improved the quality and reliability of this manuscript. I would like to make a couple of subtle remarks that do not necessarily affect the final conclusion of this work.

Response: We thank the reviewer for his positive assessment of our revision.

1. I agree that particle deformation can increase the contact area between NPs and hydrogel, but the increase in the contact area yields elevated adhesion only if the particle surface is inherently adhesive to a certain degree. If the particle surface is

completely resistant to any adhesion, deformation may not result in increased adhesion, and thus reduced diffusion rates, unless particles are physically entangled with hydrogel. As an additional possibility, particle deformation that occurs during their transport in hydrogel may lead to alteration in the surface property that renders particle more adhesive than its original state.

Response: We thank the reviewer for this insightful comment. We have revised the sentence in the revised manuscript to include the possibility of physical entanglement: *“We observed that during transportation, due to their low rigidity, the soft NPs are apt to undergo deformation which could increase the contact area between NPs and hydrogel. As a result, these NPs would suffer from the physical entanglement or increased adhesion, which led to the lowest diffusivity.”*

2. The authors made a statement that extremely small NPs are inefficiently endocytosed, which I believe is an over-generalization and the dependence of particle size on endocytosis varies with type of cells and range of particle sizes (i.e. how “extremely” small?). Prior reports demonstrate that ~100 nm NPs produced to avoid adhesion to mucus or other biological hydrogels are small enough to pass through hydrogel pores and these particles are certainly in a good range for efficient endocytosis.

Response: We thank the reviewer for his constructive comments that the demonstrations and conclusions drawn based on present data should be more specific and accurate. We have revised the sentence in the revised manuscript to include the possibility of extremely small NPs entering the cell: *“extremely small NPs (2-10 nm) are inefficiently endocytosed.”*

In addition, we agree with the reviewer that ~100 nm NPs produced to avoid adhesion to mucus are small enough to pass through hydrogel pores and these particles are certainly in a good range for efficient endocytosis. However, the optimal particle size favoring tumor penetration is not necessarily equivalent. It is reported that particles > 50 nm in diameter could hardly penetrate the tumor tissues owing to the density of the collagen network^{1, 2, 3}. Therefore, we explored the effect of particle elasticity on

mucosal delivery and tumor penetration.

Reviewer #3 (Remarks to the Author):

The authors addressed the main criticism raised by the reviewers and thereby substantially strengthened the manuscript. I can now recommend it for publication.

Response: We thank the reviewer for the appreciation of our work

References:

1. Ramanujan S, Pluen A, McKee TD, Brown EB, Boucher Y, Jain RK. Diffusion and convection in collagen gels: implications for transport in the tumor interstitium. *Biophys J* 2002, **83**(3): 1650-1660.
2. Cabral H, Matsumoto Y, Mizuno K, Chen Q, Murakami M, Kimura M, *et al.* Accumulation of sub-100 nm polymeric micelles in poorly permeable tumours depends on size. *Nat Nanotechnol* 2011, **6**(12): 815-823.
3. Lee H, Fonge H, Hoang B, Reilly RM, Allen C. The effects of particle size and molecular targeting on the intratumoral and subcellular distribution of polymeric nanoparticles. *Mol Pharm* 2010, **7**(4): 1195-1208.